# Emergent Orientation Maps —— Mechanisms, Coding Efficiency and Robustness

**Haixin Zhong**[1,2]
hxzhong@fudan

**Haoyu Wang**[3]
haoyuwang18@fudan

**Wei P. Dai**[1,4]
weidai@fudan

**Yuchao Huang**[5,6]
hyc21@mails.tsinghua

**Mingyi Huang**[1,3]
myhuang20@fudan

**Rubin Wang**[7]
rbwang@163.com

**Anna Wang Roe**[8]
annawang@zju

**Yuguo Yu**[1,2,3,4,*]
yuyuguo@fudan

1. Research Institute of Intelligent Complex Systems, Fudan University.
2. State Key Laboratory of Medical Neurobiology and MOE Frontiers Center for Brain Science, Institutes of Brain Science, Fudan University.
3. Institute of Science and Technology for Brain-Inspired Intelligence, Fudan University.
4. Shanghai Artificial Intelligence Laboratory.
5. IDG/McGovern Institute for Brain Research, School of Medicine, Tsinghua University.
6. Tsinghua-Peking Joint Center for Life Sciences.
7. Institute for Cognitive Neurodynamics, East China University of Science and Technology.
8. MOE Frontier Science Center for Brain Science and Brain-machine Integration, School of Brain Science and Brain Medicine, Key Laboratory of Biomedical Engineering of Ministry of Education, College of Biomedical Engineering and Instrument Science, Zhejiang University.
∗ Corresponding author.
Note: Append .edu.cn to author email addresses.

## Abstract

Extensive experimental studies have shown that in lower mammals, neuronal orientation preference in the primary visual cortex is organized in disordered "salt-and-pepper" organizations. In contrast, higher-order mammals display a continuous variation in orientation preference, forming pinwheel-like structures. Despite these observations, the spiking mechanisms underlying the emergence of these distinct topological structures and their functional roles in visual processing remain poorly understood. To address this, we developed a self-evolving spiking neural network model with Hebbian plasticity, trained using physiological parameters characteristic of rodents, cats, and primates, including retinotopy, neuronal morphology, and connectivity patterns. Our results identify critical factors, such as the degree of input visual field overlap, neuronal connection range, and the balance between localized connectivity and long-range competition, that determine the emergence of either salt-and-pepper or pinwheel-like topologies. Furthermore, we demonstrate that pinwheel structures exhibit lower wiring costs and enhanced sparse coding capabilities compared to salt-and-pepper organizations. They also maintain greater coding robustness against noise in naturalistic visual stimuli. These findings suggest that such topological structures confer significant computational advantages in visual processing and highlight their potential application in the design of brain-inspired deep learning networks and algorithms.

## 1 Introduction

In the primary visual cortex (V1) of visually-advanced mammals, seminal works (Hubel & Wiesel, 1962; 1974; Blasdel & Salama, 1986; Bonhoeffer & Grinvald, 1991) revealed that neurons are organized into distinct spatial clusters, forming "pinwheel" structures centered around singularities

within the orientation preference map. This organization contrasts with that of some mammals, such as rodents, which exhibit "salt-and-pepper" organizations, characterized by randomly interspersed neurons with varying orientation preferences, or "mini-columns" that show only weak spatial clustering, revealed by studies(Girman et al., 1999; Ringach et al., 2016; Bargmann & Newsome, 2014; Kaschube et al., 2010). Despite these structural differences, the spiking mechanisms that give rise to these species-specific patterns, and their roles in visual processing, remain poorly understood.

Existing computational models, such as self-organizing maps (Kohonen, 1982), on-off theoretical models (Miller, 1994; Jang et al., 2020; Song et al., 2021; Najafian et al., 2022), and various artificial neural networks (Margalit et al., 2023; Chizhov & Graham, 2021) have provided insights into the generation of orientation preference maps through mathematical abstractions or firing rate models. However, these approaches often lack the biological realism of spiking neural networks (SNNs), which better capture the intricate temporal dynamics of real neuronal systems. Furthermore, many existing SNNs (Lufkin et al., 2022; Chizhov & Graham, 2021; Srinivasa & Jiang, 2013) struggle to reproduce the emergent properties of orientation preference maps over time, as they often overlook key anatomical and physiological factors, such as shared visual inputs (Zhang et al., 2018; Jang et al., 2020; Najafian et al., 2022), cortical size (Meng et al., 2012), localized neuronal densities (Weigand et al., 2017), and genetically encoded connectivity patterns (Blauch et al., 2022).

In response to these challenges, our research contributes the following:

- We propose a retina-V1 realistic self-evolving spiking neural network (SESNN) that integrates comprehensive data on retinotopy, neuronal morphology, and connectivity from rats, cats and primates. This model is capable of dynamically simulating cortical structure formation across species using Hebbian-like learning rules and natural scene inputs. By leveraging visual field overlap and neuronal density, the SESNN accurately distinguishes between salt-and-pepper and pinwheel cortical patterns.

- Pinwheel structures arise from local synaptic plasticity rules in spiking neural networks, where lateral connections are strengthened between neurons with similar stimulus responses and weakened between those with dissimilar responses. This selective connectivity forms iso-orientation domains with consistent orientation tuning. Near pinwheel centers, connections become more homogeneous, while neurons maintain diverse orientation preferences and dynamic firing patterns. This balance between connectivity and neural diversity enables efficient, sparse coding of both simple and complex visual stimuli, optimizing visual information processing.

- We further demonstrate that pinwheel organizations exhibit superior information-cost efficiency, increased coding sparsity, and greater robustness to noise compared to salt-and-pepper patterns. These properties enhance the detection and coding of varied orientation features, indicating that the spatial structure of pinwheels promotes highly efficient and reliable visual processing.

## 2 RETINA-V1 TOPOLOGICAL SELF-EVOLVING SPIKING NEURAL NETWORK

Table 1: SESNN pinwheels vs. macaque pinwheels. (mean ± SD)

| Metric | E-I baseline | SESNN (macaque) | Macaque |
|---|---|---|---|
| Pinwheel density (pinwheels/$\Lambda^2$) | $\sim 2.941$ | $3.175 \pm 0.397$ | $\sim 3.327$ |
| Nearest-neighbor pinwheel distance (NNPD) (mm) | N/A | $0.277 \pm 0.043$ | $\sim 0.242$ |
| Hypercolumn size (mm) | N/A | $0.839 \pm 0.054$ | $\sim 0.760$ |

### 2.1 STIMULUS TRAINING SETS

We use 20 original whitened images which reduce pairwise correlations among pixels, mimicking the processing of the lateral geniculate nucleus (Dan et al. (1996); Zylberberg et al. (2011); Paul D. King et al. (2013)). To ensure these images encompassed orientation details, each image is rotated clockwise every 90-degree clockwise rotation and a subsequent flip, producing 8 variations per

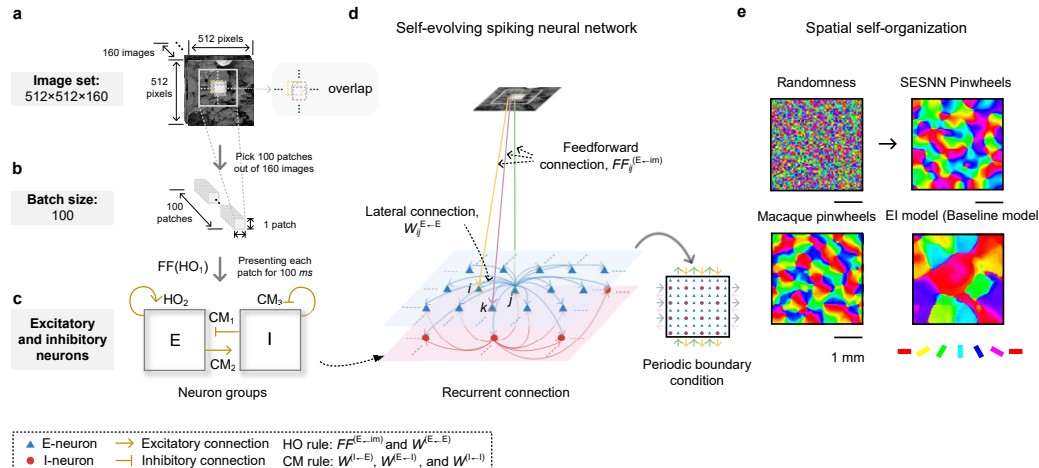

Figure 1: Architecture of proposed SESNN model. **a**. The image set with 160 whitened natural images. **b**. Stimulus images are randomly selected from the image set and presented to E neurons for 100 ms. **c**. The connections between excitatory (E) and inhibitory (I) neurons follow HO and CM rules. **d**. Recurrent connection among neurons. The spatial arrangement and periodic boundary conditions of E- and I- neurons. E- and I- neurons share the same spatial coordinates, and the boundaries are linked together according to the arrows shown in the diagram. Arrows of the same color represent identical connections. **e**. Spatial self-organization within the neuronal population post-training and the comparison among the SESNN model's orientation preference map, macaque's orientation preference map and SNN-based model (baseline) (Srinivasa & Jiang, 2013). The color bars represent orientation preferences, and the scale bar corresponds to 1 mm on the V1 cortical surface. This comparative analysis is detailed in Table 1.

original image. This method generates a total of 160 images comprising our training dataset (refer to Fig. 1a).

In each trial, all E-neurons process visual input from 100 random patches, each presented for 100 ms(see Fig. 1b). Each E-neuron perceives a small patch of 16×16. The visual fields of the E-neurons are modeled as overlapping on the retina (see Fig. 1a), with the total visual input dimension being $[(16 - \varepsilon)\sqrt{n} + \varepsilon]$ pixels on each side ($n$ is neuron number, visual overlap metric $\varepsilon$). To address the visual overlap among adjacent neurons, which adheres to periodic boundary conditions, the input design features a wraparound effect to ensure continuity and uniformity throughout the neural network.

## 2.2 NEURAL MODEL AND THE PLASTICITY RULES

The SESNN model includes 4900 E- and 1225 I- neurons, forming a recurrent network with an E-I ratio of 4:1 (Alreja et al. (2022); Tao et al. (2004); Pfeffer et al. (2013)). E-neurons are stimulated by natural images and receive background noise from other brain areas, while I-neurons indirectly process natural image stimuli through their interactions with E-neurons. The spiking dynamics of the network are simulated by the classical leaky integrate-and-fire neurons (Stevens et al. (2013)), which also feature refractory periods of 3 ms and adaptive firing thresholds (Földiák (1990)). The detailed description of the neuronal model can be found in the appendix (A.1).

The learning process utilizes two primary forms of Hebbian-like learning rules: Hebbian Oja's variant (HO) and Correlation Measuring (CM) (Oja (1982); Zylberberg et al. (2011); Paul D. King et al. (2013)). Consistent with experimental evidence Holmgren et al. (2003); Hofer et al. (2011), V1 pyramidal neurons exhibit weaker synaptic strengths compared to other types, crucial for avoiding over-excitation and ensuring neural balance. To accurately model these self-evolving dynamics within our SESNN, we implement E-E connections governed by the HO rule (Oja (1982)), with a normalization factor (the second term in Eq. 1) to regulate synaptic weight growth within a 0 to 1 range. In contrast, lateral connections between excitatory and inhibitory neurons are governed by the CM

rule, which inherently allows for stronger synaptic weights without normalization (Zylberberg et al. (2011); Paul D. King et al. (2013)), enabling these connections to better capture correlations between input patterns and neuronal responses. This distinction in learning rules is crucial for maintaining functional balance and enhancing representational efficiency within the network. Specifically, the HO rule learns the principal component of input patterns, allowing E-E connections to align with the largest variance in the input space. While this supports self-organized structure formation, such as oriented receptive fields (RFs), it can lead to redundancy across excitatory and inhibitory neurons. By comparison, the CM rule captures correlations between input activity and a neuron's response, ensuring that predictable components of excitatory activity are expressed in inhibitory neurons. These components can then be suppressed through E-I-E loops, a mechanism that supports predictive coding by minimizing redundancy and promoting sparse, efficient neural representations. Both HO and CM rules support long-term potentiation and depression, functioning similarly to other rate-based learning rules that do not require precise spike timing. We selected the HO and CM rules for their minimal tuning requirements and their ability to stabilize recurrent excitation.

Following every 100 ms of stimulus presentation, synaptic weight changes $\Delta W$ within the network are computed:

$$\text{HO}: \ \Delta W_{ij}^{(\text{K} \leftarrow \text{K}^*)} \propto y_i x_j - y_i^2 W_{ij}^{(\text{K} \leftarrow \text{K}^*)}, \tag{1}$$

$$\text{CM}: \ \Delta W_{ij}^{(\text{K} \leftarrow \text{K}^*)} \propto y_i x_j - \langle y_i \rangle \langle x_j \rangle \left( 1 + W_{ij}^{(\text{K} \leftarrow \text{K}^*)} \right), \tag{2}$$

where $x$, $y$ refer to the spike rate of presynaptic ($j$) and postsynaptic neurons ($i$) and K, $\text{K}^*$ determine the type of neurons (E or I), the operator $\langle \cdot \rangle$ indicates the lifetime average value. Following each 100 ms stimulus presentation, we compute the network's neuronal instantaneous spike rates as exponential moving averages, which accumulate spikes over time (see Eq. 3). Exponential moving averages are utilized to track recent neuronal activity levels. Concurrently, lifetime average values are also calculated using exponential moving averages, which are crucial for maintaining homeostatic stability. This method helps stabilize the neural network by adjusting neuronal properties or synaptic strengths to sustain consistent activity levels over time.

$$x_j(t) = (1 - \zeta) x_j(t - 1) + \zeta \cdot z_j(t), \tag{3}$$

where $\zeta = 1 - e^{-\frac{1}{10}}$, indicating that the 10 ms is a temporal window of the moving average weighted with exponential decay. The initialization of $x_j$ is 0. The exponential moving average is calculated dynamically and updated along with synaptic weights. $z_j^{(\text{K}^*)}(t) = 1$ represents the spike output from neuron $j$ at time $t$, and 0 otherwise.

$$\langle x_j \rangle := (1 - \xi) \cdot \langle x_j \rangle + \xi \cdot \overline{x}_j, \tag{4}$$

where $\xi = 1 - e^{-1}$. It is dynamically updated to ensure the sum of the weights remains constant over time. Eq. 3 captures the short-term firing rate of neurons by tracking recent activity through a short time window, while Eq. 4 computes the long-term average to monitor activity trends and maintain homeostatic stability, together ensuring the network's adaptability and balanced dynamics over multiple timescales. $\overline{x}_j$ represents the average firing rate of neuron $j$ in one trial. The hyperparameters of the SESNN model are as follows: the learning rates are $\eta_{\text{FF}} = 0.2$ (image to E-neuron), $\eta_{\text{EE}} = 0.01$ (E- to E-neuron), $\eta_{\text{EI}} = 0.7$ (I- to E-neuron), $\eta_{\text{II}} = 1.5$ (I- to I-neuron), and $\eta_{\text{IE}} = 0.7$ (E- to I-neuron), while the neural connectivity parameters are $\alpha_{\text{max, E}} = 1.0$ (E- max weight) and $\alpha_{\text{max, I}} = 0.5$ (I- max weight). These learning rate settings are crucial for stabilizing the training of the neural network. Specifically, setting a slower learning rate for E-E connections than for others helps prevent over-excitation among E-neurons. This approach is consistent with empirical findings (Hofer et al., 2011; Holmgren et al., 2003; Sato et al., 2016).

## 2.3 EXPERIMENTAL DATA-JUSTIFIED NEURAL CONNECTIVITY IN A 2D CORTICAL AREA

E- and I- neurons are arranged symmetrically on a two-dimensional lattice within SESNN. The initial connection weights between neurons are modeled by a Gaussian function of Euclidean distance, which can be expressed as:

$$W_0^{\text{K} \leftarrow \text{K}'}(i, j) = \alpha_{\text{K}'} \times \exp\left( \frac{-d(i, j)^2}{2\sigma_{\text{K}'}^2} \right), \tag{5}$$

where $d(i, j)$ represents the Euclidean distance from neuron $i$ to neuron $j$ in a grid, $\alpha$ determines the maximum connection weight, which is set to $\alpha_{EE} = 1$, $\alpha_{EI} = 1$, $\alpha_{IE} = 0.5$, $\alpha_{II} = 0.5$, and $\sigma$ governs the rate at which the weight decays with distance. The synaptic types predominantly determine the parameters for this connection weight distribution function. To accurately replicate the neuronal architecture of V1 in macaques. The synaptic connection range, denoted by $\sigma$, are set to $\sigma_{EE} = 3.5$, $\sigma_{EI} = 2.9$, $\sigma_{IE} = 2.6$, $\sigma_{II} = 2.1$. These values are based on anatomical data indicating that the axon lengths of E- and I-neurons are approximately 200 $\mu$m and 100 $\mu$m, respectively, while the dendrite lengths are around 150 $\mu$m for E-neurons and 75 $\mu$m for I-neurons in the V1 (Tao et al. (2004); Stepanyants et al. (2009); Amatrudo et al. (2012)). This careful selection of parameters and neuron distribution ensures our model's biological fidelity, particularly in representing synaptic connectivity and neuronal density. We discard any connection strengths below a threshold of 0.01 to maintain computational efficiency and biological plausibility.

Given that the majority of RFs are concentrated in the area centralis Van Essen et al. (1984); Born et al. (2015), which projects to a disproportionately large area of V1, we focus our analysis solely on RFs within this central visual field (see Table 2). To control for variables, we use a constant RF size near the central vision, where eccentricity is close to zero.

The anatomical data for neural connectivity is based on the findings of (Tao et al., 2004; Stepanyants et al., 2009; Amatrudo et al., 2012). The plasticity rule settings are based on the findings of (Holmgren et al., 2003; Hofer et al., 2011). For retinotopic data, we refer to (Srinivasan et al., 2015; Tehovnik & Slocum, 2007; Scholl et al., 2013; Veit et al., 2014; Engelmann & Peichl, 1996; Weigand et al., 2017; Law et al., 1988; Huberman et al., 2006; Niell & Stryker, 2008; van Beest et al., 2021; Foik et al., 2020). The anatomical data on orientation preference maps is sourced from (Najafian et al., 2022). The details of the data usage and visual input overlap calculation can be found in the appendix A.2,A.3,A.4.

Table 2: Comparative anatomical data of the retina and V1 across species.

| Species | V1 neurons density (neurons/mm$^2$) | V1 RF size (deg) | Peak CMF (mm/deg) | Pinwheel numbers (per mm$^2$) |
|---|---|---|---|---|
| Macaque | 243,000[a] | 0.2[b] | 18.18[b] | 8 |
| Cat | 99,200[a] | 1.0[c] | 1.90[d] | 3 |
| Mouse | 86,600[a] | 4.0[e] | 0.03[f] | N/A |

[a] Srinivasan et al. (2015) [b] Tehovnik & Slocum (2007) [c] Scholl et al. (2013)
[d] Tusa et al. (1978) [e] Niell & Stryker (2008) [f] van Beest et al. (2021)

## 3 SESNN REVEALS THAT VISUAL OVERLAP BETWEEN NEURONS IS CRITICAL FOR V1 ORIENTATION MAP FORMATION

For quantitative analysis of orientation maps, we utilize metrics including pinwheel density (pinwheels/mm$^2$), pinwheel counts (Kaschube et al. (2010); Stevens et al. (2013)) (Fig. 2b), NNPD (mm) (Müller et al. (2000); Kaschube et al. (2010)) (Fig. 2c), and hypercolumn size (mm) (Stevens et al. (2013)) (Fig. 2d). Pinwheel density is defined as the number of pinwheels per V1 surface area mm$^2$. NNPD refers to the shortest Euclidean distance between pinwheel centers. Pinwheel centers are identifiable via 2D Fast Fourier Transform (Kaschube et al. (2010)), located where the real and imaginary components of the transform intersect at zero (Stevens et al. (2013)). A hypercolumn is defined as a region with periodicity measured by 2D fast Fourier transform. It's noteworthy that pinwheel density is excluded from the Fig. 2 analysis, because, regardless of hypercolumn size, the observed pinwheel density consistently approaches $\pi$ pinwheels/$\Lambda^2$(Fig. 3d), conforming to topological constraints (Stevens et al. (2013); Kaschube et al. (2010)). This paper does not account for left- and right-eye dominance columns, so the hypercolumn size is defined as the full 180° cycle of repeating column spacing ($\Lambda$) (mm).

Our SESNN model effectively generates various orientation selectivity patterns, from salt-and-peppers to pinwheel structures. We modify the overlap parameter $\varepsilon$ to range from 10 to 15 pixels (see Fig. 2). This visualization highlights how variations in the degree of overlap in visual stimuli can influence

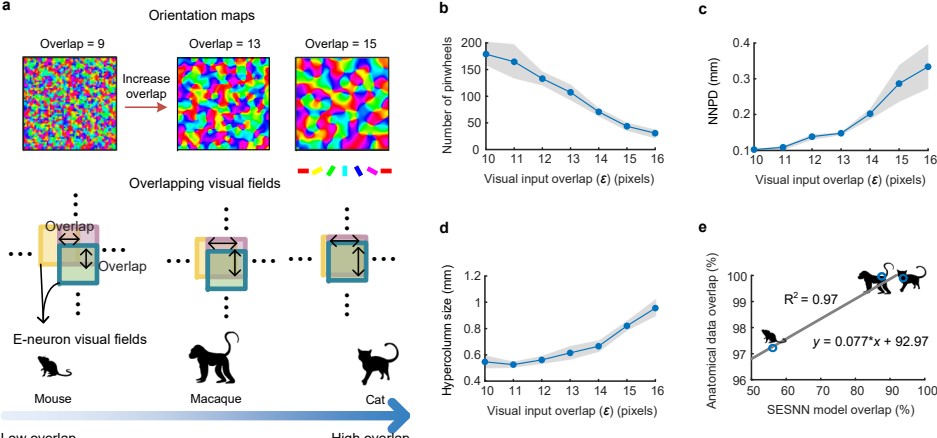

Figure 2: The overlap of visual fields among adjacent neurons contributes to the formation of pinwheel structures. **a**. The diverse overlapping extent of visual fields in neurons forms various iso-orientation domains of species in the orientation preference maps when perceiving natural images. The increasing overlap is consistent with the transition from lower to higher mammals (see the animal sketches). **b**. The correlation between visual overlap and pinwheel counts. **c**. The relationship between visual overlap and NNPD. **d**. The hypercolumn size varies with different visual overlaps. **e**. Comparing the SESNN model overlap with actual anatomical data overlap percentages in different species (mice, cats, and macaques). Color scheme: orientation preference. All data: mean ± SD.

the density and arrangement of pinwheel structures in the visual cortex. Notably, setting the overlap to $\varepsilon_2 = 9$ pixels results in the model producing salt-and-pepper organizations (see the first panel of Fig. 2a), rather than the anticipated pinwheel structures. Thus, it is evident that above the threshold of overlap (in our scenarios, $threshold_\varepsilon = 10$ pixels), pinwheel structures emerge. Specifically, we find that a greater overlap (e.g., $\varepsilon_2 = 15$ pixels) fosters the formation of stronger local clusters, resulting in larger hypercolumn sizes (Fig. 2d), a reduced number of pinwheels (Fig. 2b), and extended NNPD (Fig. 2c), compared with the scenario of a smaller overlap (e.g., $\varepsilon_1 = 12$ pixels). These findings suggest that visual input overlap critically determines the species-specific organization patterns of V1. The overlap degrees ($\varepsilon$) in visual fields vary across species (including macaques, cats, rodents, etc.) (Fig. 2e) to accommodate the size of iso-orientation domains in the visual cortex representation as observed by the study of (Najafian et al., 2022). When the overlap of visual stimuli is large, multiple neurons receive similar visual inputs. This causes these neurons to form stronger local connections through Hebbian plasticity. Neurons with similar RFs tend to develop highly consistent local connections, and their orientation preferences gradually align, creating a region of locally continuous orientation preferences. Additionally, Neuronal density is another factor that impacts the size of hypercolumns. Although this has been reported previously Philips et al. (2017), we have included additional details in Appendix A.3.2 for completeness.

## 3.1 THE EMERGENCE OF THE PINWHEEL STRUCTURE AND MECHANISM

We investigate the development of orientation preference maps in V1 of animals at eye-opening using the SESNN model, where the initial visual experience of natural stimuli are retinotopically mapped to V1 neurons (Fig. 3a) (Stevens et al. (2013)). Our comparative analysis with empirical ferret data (Stevens et al. (2013); Chapman et al. (1996); Chapman & Stryker (1993)) shows a high correlation ($r = 0.98$, Pearson correlation) for orientation preference maps observed from postnatal day 33 (P33) to P41 when they reach high stability (Fig. 3b). By normalizing the time scale, we are able to align the SESNN model's developmental trajectory with ferret empirical data. To quantitatively assess how the stability change over time, we use the orientation stability index (SI) (Chapman et al. (1996)), calculated as follows:

$$\text{SI} = 1 - \frac{4}{n\pi} \sum_{i=1}^{n} \left| (F_i - O_i) \bmod \left( \frac{\pi}{2} \right) \right|. \tag{6}$$

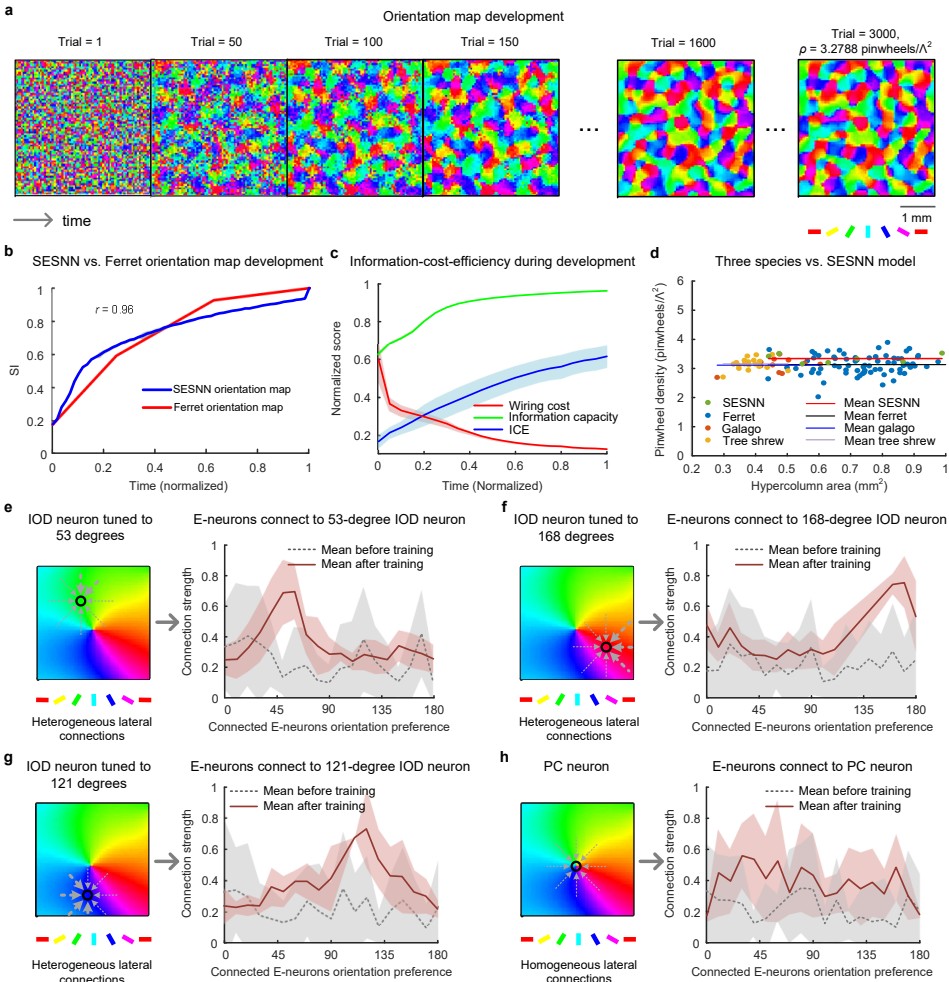

Figure 3: Orientation preference map development in SESNN model. **a**. The temporal evolution of the SESNN model's orientation preference maps, presented in HSV format. The final trial orientation map displays measured density per hypercolumn area ($\Lambda^2$). **b**. The correlation between orientation preference maps from the SESNN model and observed orientation preference maps in the ferret over a normalized time scale. **c**. The changes in three different metrics—wiring cost, information capacity, and information-cost-efficiency during orientation map development. **d**. Comparative analysis of pinwheel densities relative to hypercolumn areas for ferrets, galagos, and tree shrews (Kaschube et al. (2010); Stevens et al. (2013)), alongside the SESNN model. **e-h**. Subplots with insets of pinwheel structures identify focal neurons and depict changes in connection strength among E-neurons with various orientation tunings before and after training (dashed and solid lines: mean). **e-g**. Display heterogeneous connection weights that bias toward iso-orientation domains. **h**. The subplot shows that the homogeneous connection weights contributed equally from nearby iso-orientation domains. The thickness of the arrows represents connection strength. All shaded areas: SD.

The SI is computed by iterating over all $n$ data points, where $i$ indexes individual pixels, $F_i$ represents the final orientation preference value of the neurons, and $O_i$ denotes the orientation preference at a prior developmental stage. The higher SI at later development stage of the orientation preference maps implying that maps remain unchanged during their continued exposure to varying natural stimuli (Fig. 3b). Furthermore, we also investigated whether the model's performance would change if trained with a new set of images after stabilization. Our results show that while some detailed aspects of the pinwheel structure altered slightly, the overall locations of the pinwheel centers and the domains remained largely unchanged, indicating strong stability. The changes in neuron tuning toward specific orientations were also observed (detailed in Appendix A.5). Additionally, we examine

the relationship between dimensionless pinwheel densities ($\rho$) and hypercolumn areas ($\Lambda^2$) across the SESNN model, ferrets, galagos, and tree shrews, comparing findings with empirical data (Fig. 3d).

Considering the trained connections $W_{ij}$, we follow the hypothesis that the information capacity can be depicted by the entropy of weight distribution in the complete connection matrix (distribution of all the trained connection strength) and the energy cost of synaptic transmission is inversely proportional to the connection strength. We have the information-cost efficiency (ICE) defined as:

$$\text{ICE} = \frac{H}{C}; H = -\sum_k P_k \log\left(P_k\right); C = \sum_{i,j} \frac{1}{W_{ij}}, \tag{7}$$

where $H$ represents the information encoding capacity, and $C$ represents the energy cost of synaptic transmission, $P_k$ is the discrete probability of the distribution, that is, the normalized count of $W_{ij}$ in k-bins of the same size (0.01). Here, the calculation of wiring costs is based on Eq. 7 from Bertens & Lee (2022). In this approach, the network wiring cost is inversely proportional to the summed synaptic connection weights between pairs of neurons, indicating that stronger synaptic weights not only facilitate efficient communication but also contribute to lower wiring costs. In Fig. 3c, we observed decreasing wiring costs during the development, which is consistent with previous research (Koulakov & Chklovskii, 2001). Meanwhile, the growing information capacity and the ICE suggest that the emergence of the pinwheel structure effectively maximizes coding capacity under the constraint of energy cost.

To further understand the circuit mechanisms underlying the emergence of pinwheel structure, we examine the connections between E-neurons for ones at the iso-orientation domains (e.g., Fig. 3e-g) and ones at the pinwheel centers (e.g., Fig. 3h). All connections of E-neurons are initialized with Gaussian-distributed strengths (see Eq. 5) and distributed uniformly across neurons of different orientation preferences (determined after stabilization), as indicated by the gray dashed curves. After the orientation preference maps stabilized, the connectivity patterns evolved significantly, leading to selectively stronger connections to neurons with similar orientation preferences. This is clearly demonstrated by the red solid curves and the varying thickness of the arrows. The temporal in-variance of the connection distribution at the pinwheel center over time (except for an increase in overall strength), therefore, clearly demonstrated that the iso-orientation domains must take form first, as also visually verified in early trials of our model (Fig. 3a). This local continuity of iso-orientation domains is consistent with the demand of Hebbian-like learning rule (Eq. 1) together with the overlap between RFs. The domain size is determined by the extent of E-connections, while swirling patterns arise from the varied local orientations in stimuli, enabling animals to perceive different orientations within the same visual field while maintaining local continuity. In salt-and-pepper organizations, reduced overlap disrupts this continuity. Thus, visual stimulus overlap shapes orientation preference maps in V1: greater overlap fosters pinwheel structures, while reduced overlap leads to salt-and-pepper patterns.

## 3.2 CODING EFFICIENCY AND ROBUSTNESS EMERGENCE WITHIN PINWHEELS

To assess the response reliability of neurons under stochastic conditions—vital for trustworthy and robust neural information processing—we analyze neuron reliability within orientation preference maps under noisy environments ($\mathcal{N}(0, \sigma_r^2)$). We utilize a spike-based measurement approach to quantify this reliability. Specifically, for each neuron, we examine spike time sequences generated under identical stimulus conditions with added noise, calculating the mean cross-correlation between these sequences. We then determine the neuron's reliability by identifying the maximum in the average cross-correlation. For each neuron, we analyze $n$ spike trains of duration $T$. After normalizing the mean of each train to zero, we systematically calculate the cross-correlation coefficient between each pair of sequences from these $n$ trains:

$$R_{xy}(m) = E\left[x_{T+m} y_T\right], \tag{8}$$

where $m$ denotes lag, the range of which can extend over the length of the time sequence $T$. $x$ and $y$ denote two different spike trains. Through this procedure, the average correlation coefficient of the same neuron can be determined:

$$\overline{R}_{xy}(m) = \frac{\sum_{i=1}^{C_n^2} R_{xy}(m)}{C_n^2}. \tag{9}$$

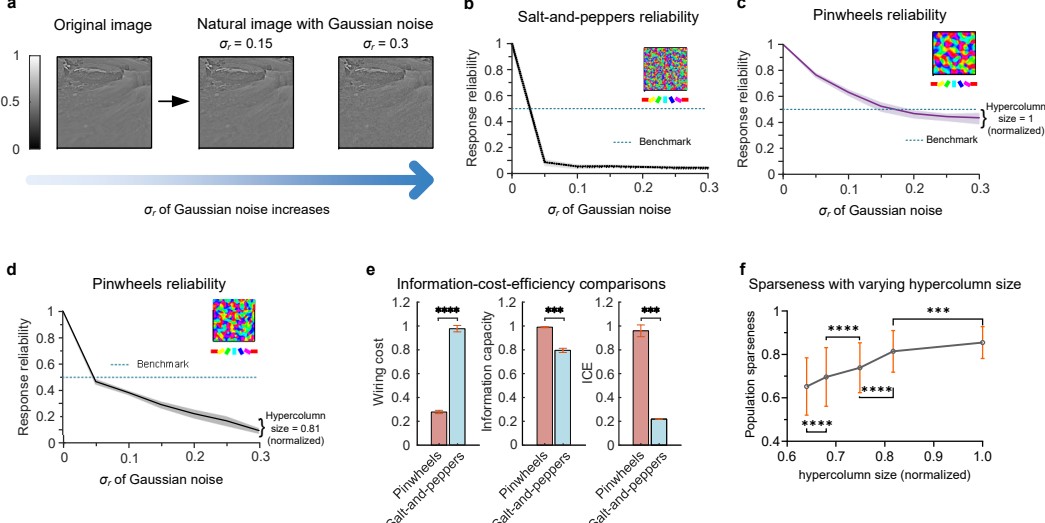

Figure 4: Population response analysis investigates neurons across different orientation preference maps. **a**. The SESNN model is trained using natural image inputs that include background disturbances characterized by Gaussian noise ($\sigma_r \in [0, 0.3]$). **b**. The response reliability of all neurons in the salt-and-peppers gradually decreases as $\sigma_r$ increases. **c-d**. The response reliability about noise level decreases distinctly for neurons within the pinwheel center and surrounding areas across different hypercolumn sizes. Insets illustrate these patterns, with color bars indicating orientation preferences. A dashed blue line at $y = 0.5$ acts as a benchmark for comparing the responses of neuron types under specific noise conditions. **e**. Comparisons of three metrics—wiring cost, information capacity, and ICE—across between pinwheels and salt-and-peppers. **f**. The averaged population response sparseness of neurons increases for pinwheels with larger hypercolumn sizes. (Data presented as mean ± SD, statistical significance denoted as ***p<0.001, ****p<0.0001, Welch's *t*-test).

We posit that the maximum lag for similar time sequences generated by the same neuron cannot exceed half the length of the time sequence. Consequently, the reliability is obtained as follows:

$$\text{Reliability} = \max_{-\frac{T}{2} \le m \le \frac{T}{2}} \overline{R}_{xy}(m). \tag{10}$$

To assess the response reliability of the SESNN model, we use a noise-corrupted natural image as the input. This involves superimposing Gaussian white noise onto the image. The model is then stimulated with this noise-modified image, with the same variance noise applied in 10 trials.

Our findings show that neurons in orientation maps with pinwheel structures are more reliable than those in salt-and-pepper or mini-column configurations (Fig. 4b-d). Pinwheel structures in higher mammals (Fig. 4c-d) enhance signal processing reliability against noise and visual perception. In Fig. 4e, we compare wiring cost, information capacity, and ICE across pinwheels and salt-and-peppers. Pinwheels exhibit lower wiring costs and higher information capacity, making them more efficient, as indicated by the ICE metric (Crumiller et al., 2013). Additionally, we calculate the population sparseness (Vinje & Gallant, 2000) for different orientation map sizes using 100 natural image patches within a 100 ms window (Fig. 4f). Population sparseness measures the distribution of neuron responses to a single patch, as depicted below:

$$\text{Population sparseness} = \frac{1 - \left( \sum \frac{\langle a_i \rangle}{m} \right)^2 / \sum \left( \frac{\langle a_i \rangle^2}{m} \right)}{1 - \frac{1}{m}}, \tag{11}$$

where $a_i$ refers to the mean spike rate of a neuron $i$ out of a total of $m$ neurons in a time window.

The model's findings reveal a pattern in which population sparseness intensifies with the enlargement of hypercolumn sizes, spanning from small to large across various species. This pattern is illustrated in Fig. 4f, which displays the progressively rising average population sparseness about hypercolumn size

across 100 patches. Our model's findings are consistent with empirical data (population sparseness: SESNN pinwheels $\sim 0.74$ vs. $\sim 0.67$ macaque pinwheels (Zandvakili & Kohn, 2015), SESNN salt-and-pepper organizations $\sim 0.25$ vs. $\sim 0.36$ mice (Froudarakis et al., 2014).

## 4 RELATED WORKS

**Traditional models.** Traditional models such as self-organizing maps (Kohonen, 1982), ON-OFF theoretical models (Miller, 1994; Jang et al., 2020; Song et al., 2021; Najafian et al., 2022), and artificial neural networks (Margalit et al., 2023; Chizhov & Graham, 2021) have explored orientation map mechanisms but often lack the complexity found in spiking neural networks. Recent spiking neural networks (Lufkin et al., 2022; Keane & Gong, 2015; Chizhov & Graham, 2021) have struggled with static inputs and connections, failing to accurately capture neural dynamics. Most importantly, no previous spiking neuronal networks with plasticity have actually successfully reproduced pinwheel structures. Line discontinuity instead discrete point discontinuity (as in real pinwheel and our model) plague the functional map (Srinivasa & Jiang, 2013). Thus, as a major improvement, our model's quantitative match with experiment results greatly enhanced the explaining power of its mechanisms.

**Visual overlap among neighboring neurons.** Numerous studies have explored the variability in V1 neuronal organization and its impact on afferent segregation, with factors like visual sampling density playing a significant role (Baden et al., 2016; Román Rosón et al., 2019; Srinivasan et al., 2015; Najafian et al., 2022; Jin et al., 2011; Lee et al., 2016; Kremkow et al., 2016; Jang et al., 2020; Lien & Scanziani, 2013). While these studies have highlighted the importance of anatomical factors such as visual overlap and neuronal density in distinguishing pinwheel structures from salt-and-pepper organizations, they primarily rely on phenomenological static models, lacking detailed mechanisms for how these factors influence functional organization. Our study provides a clear mechanism by demonstrating that visual input overlap is crucial in forming pinwheel structures. Through our spiking neural network model, we show that increased overlap between neighboring neurons enhances local connectivity and functional clustering in iso-orientation domains, leading to the emergence of pinwheels. In contrast, reduced overlap weakens these local connections, resulting in the disorganized salt-and-pepper patterns. This finding establishes a direct link between the structure of input overlap and the functional organization of V1, making the connection between structure and function explicit.

**Wiring cost minimization and sparse coding strategy.** Research suggests that wiring minimization optimizes orientation maps (Koulakov & Chklovskii, 2001; Petitot, 2017; Dai et al., 2018). Our study expands on this by not only examining the wiring costs but also the information capacity, thereby finding that the ICE of pinwheel structures is much improved compared to salt-and-pepper organizations. Studies also suggest that sparseness in neural coding enhances precision and reduces redundancy (Zhou & Yu, 2018; Yu et al., 2014; Olshausen & Field, 1997; Rolls & Tovee, 1995; Vinje & Gallant, 2000). We found that pinwheel structures strike a balance between reliability and coding sparsity. Lateral connections in pinwheels strengthen synapses among neurons with similar orientations and weaken those with dissimilar orientations. Thus, by enhancing functional clustering in the iso-orientation domains, these neurons boost reliability with some redundancy but still effectively implement sparse coding by suppressing nearby iso-orientation domains. While near pinwheel centers, where connections are less selective, neurons display varied response patterns instead of clear orientation preferences, therefore implements sparse coding of high-order pattern in addition to plain feature space of single orientations.

## 5 CONCLUSION

The SESNN model integrates anatomical and physiological factors like visual overlap, neuronal density, and synaptic plasticity, providing a more realistic, dynamic representation of spiking mechanisms in V1, especially in pinwheel formation. Our study identifies visual overlap, neuron density, and local plasticity as key factors driving the transition from salt-and-pepper to pinwheel topologies, linking structural features to functional performance—a significant advancement in understanding visual coding efficiency. By addressing wiring cost minimization and information capacity, our findings reveal optimal cortical network strategies for sparse and robust visual encoding. These insights can inform the design of more efficient brain-inspired computational models.

## ACKNOWLEDGMENTS

We extend our sincere gratitude for the support provided by the Science and Technology Innovation 2030 - Brain Science and Brain-Inspired Intelligence Project (2021ZD0201301), as well as the National Natural Science Foundation of China (U20A20221, 12201125, 12072113). We also acknowledge funding from the Shanghai Municipal Science and Technology Committee of Shanghai Outstanding Academic Leaders Plan (21XD1400400), the Yang Fan Plan (22YF1403300), and the China Postdoctoral Science Foundation (2023M740724).

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

## A  APPENDIX

### A.1  DETAILED NEURAL MODEL

The spiking neuronal dynamics are iteratively formulated as follows:

$$u_i^{(K)}(t+1) = u_i^{(K)}(t)e^{-\frac{\eta}{\tau^{(K)}}} + h_K(i)\sum_j FF_{ij}^{(E\leftarrow \text{image})} X_j$$
$$+ \sum_{K^*}\sum_j \beta_{ij}^{(K\leftarrow K^*)} \cdot W_{ij}^{(K\leftarrow K^*)} \cdot z_j^{(K^*)}(t) + \text{noise}, \tag{12}$$

$$h_K(i) = \begin{cases} 1, & \text{if } i \text{ is an E-neuron ID,} \\ 0, & \text{if } i \text{ is an I-neuron ID,} \end{cases} \tag{13}$$

$$\Delta\theta_i^{(K)} \propto p_i(z_i^{(K^*)} = 1) - p_i^{(K)}, \tag{14}$$

where K = Excitatory (E) neurons or Inhibitory (I) neurons; $i = 1, 2, \ldots, N$ (numbers of E-neurons and I-neurons).

In Eq. 12, $u_i^{(K)}(t)$ represents the membrane potential of neuron $i$ at time $t$, applicable to neurons of class K. The decay rate of the membrane potential is governed by the resistor-capacitor circuit time constant $\tau$, set at $\tau^{(E)} = 10$ ms for E-neurons and $\tau^{(I)} = 5$ ms for I-neurons, reflecting the faster firing rate of I-neurons compared to E-neurons (Paul D. King et al. (2013); Thomson & Lamy (2007)). This configuration reduces reconstruction error and speeds up system convergence, enhancing the representation of input stimuli. The feedforward (FF) connection, $FF_{im}^{(E\leftarrow\text{image})}$, connects pixel $X_m$ from a whitened image patch (normalized to zero mean and unit variance) to E-neuron $i$. Connection weight $W_{ij}^{(K\leftarrow K^*)}$ from neuron $j$ of class $K^*$ to neuron $i$ of class K includes a sign $\beta_{ij}^{(K\leftarrow K^*)}$ indicating excitatory (+1) or inhibitory (-1) connections. $z_j^{(K^*)}(t) = H\left(u_j^{(K^*)}(t) - \theta_j^{(K^*)}\right)$, where $H(x)$ is the Heaviside step function, defined as $H(x) = 1$ if $x > 0$ and 0 otherwise. Here, $z_j^{(K^*)}(t)$ represents the spike output from neuron $j$ at time $t$, which triggers a reset of the membrane potential to 0 after reaching the spike threshold $\theta$ (initially set at 2). The firing threshold $\theta$ adapts based on the discrepancy between the current and target firing rates $p_i(t)$ and $p_i^{(K)}$ ($p^{(E)} = 2$, $p^{(I)} = 4$) (see Eq. 14 (Földiák, 1990)). Neurons receive Gaussian white noise $\mathcal{N}(0, 0.04)$ from other brain areas (noise term). For computational efficiency, the time step $\eta$ is set at 1 ms.

### A.2  ANATOMICAL DATA INTEGRATION

NEURAL CONNECTION DATA

The experimental subjects include six adult cats with unknown genders, with data sourced from research by Armen Stepanyants et al.(Stepanyants et al., 2009); and eight macaques, aged 5-11 years, including six males and two females, with data sourced from research by Joseph Amatrudo et al.(Amatrudo et al., 2012).

NEURONAL SYNAPTIC PLASTICITY

The subjects are rats aged 14-16 days, with unknown gender and quantity, with data sourced from research by Holmgren et al.(Holmgren et al., 2003); transgenic mice, with unknown quantity and gender, with data sourced from research by Hofer et al. (Hofer et al., 2011).

RETINAL-V1 TOPOLOGICAL PROJECTION DATA

Receptive field (RF) data: Primary visual cortex (V1) neuron counts for macaques, cats, tree shrews, ferrets, mice, rats, and gray squirrels respectively come from Tehovnik et al. (Tehovnik & Lee, 1993) (subjects: 3 macaques, unknown gender and age), Scholl et al. (Scholl et al., 2013) (subjects: cats, unknown gender and age), Veit et al.(Veit et al., 2014) (subjects: 9 male and 7 female tree shrews, aged 3-8 years), Huberman et al.(Huberman et al., 2006) (subjects: 8 ferrets, unknown gender and age), Niell et al.(Niell & Stryker, 2008) (subjects: mice, aged 2–6 months, unknown gender), Foik et

al. (Foik et al., 2020)(subjects: 21 rats, unknown gender and age), and Hall et al. (Hall et al., 1971) (subjects: 17 gray squirrels, unknown gender and age). V1 neuronal density: neuronal density data for macaques, cats, mice, rats, and gray squirrels come from Srinivasana et al. (Srinivasan et al., 2015) (subjects: unknown gender and age); tree shrew, ferret, and gray squirrel density data respectively come from Weigand et al. (Weigand et al., 2017).

CORTICAL MAGNIFICATION FACTOR

Cortical magnification factor (CMF) data for macaques, cats, tree shrews, ferrets, mice, rats, and gray squirrels are sourced from Tehovnik et al. (Tehovnik & Lee, 1993) (subjects: 3 macaques, unknown gender and age), Veit et al. (Veit et al., 2014) (subjects: cats, unknown gender and age), Bosking et al. (Bosking et al., 1997) (subjects: tree shrews, unknown gender and age), Rockland et al. (Law et al., 1988) (subjects: 9 ferrets, female, unknown age), Beest et al. (van Beest et al., 2021) (subjects: 28 mice, 11 males and 17 females, ages 2-14 months), Keller et al. (Keller et al., 2000) (subjects: male rats, age 3 months), and Hall et al. (Hall et al., 1971) (subjects: 17 gray squirrels, unknown gender and age).

### A.3 UNVEILING SPECIES-SPECIFIC FACTORS DISTINGUISHING PINWHEELS AND SALT-AND-PEPPERS

#### A.3.1 ANATOMICAL DATA SUGGESTS RFs DENSITY UNDERLYING V1 ORGANIZATIONS

Table 3: Comparative anatomical data of the retina and V1 across species.

| **a.** Species (mean) | **b.** Retina ($mm^2$) | **c.** V1 size ($mm^2$) | **d.** V1 neurons density (neurons/$mm^2$) | **e.** V1 RF size in area centralis (deg) | **f.** RFs density $((c) \times (d)/(b))$ (RFs/$mm^2$) |
|---|---|---|---|---|---|
| Macaque | $636^a$ | $1,090^b$ | $243,000^b$ | $0.2^c$ | 416,462.26 |
| Cat | $510^a$ | $380^b$ | $99,200^b$ | $1.0^d$ | 73,913.73 |
| Tree shrew | $122^{a,e}$ | $73^{b,a}$ | $192,800^f$ | $2.0^g$ | 115,363.93 |
| Ferret | $83^{h,a}$ | $78^b$ | $95,813^f$ | $3.0^i$ | 90,041.13 |
| Mouse | $15^a$ | $2.5^b$ | $86,600^b$ | $4.0^j$ | 14,433.33 |
| Rat | $52^{a,k}$ | $7.1^b$ | $90,800^b$ | $3.0^l$ | 12,397.69 |
| Gray squirrel | $205^a$ | $32^a$ | $84,213^f$ | $2.0^m$ | 13,145.44 |

[a] Jang et al. (2020) [b] Srinivasan et al. (2015) [c] Tehovnik & Slocum (2007)
[d] Scholl et al. (2013) [e] Engelmann & Peichl (1996) [f] Weigand et al. (2017)
[g] Veit et al. (2014) [h] Law et al. (1988) [i] Huberman et al. (2006)
[j] Niell & Stryker (2008) [k] Hughes (1979) [l] Foik et al. (2020)
[m] Hall et al. (1971)

We analyzed anatomical data from seven species, including primates (e.g., macaques) and non-primates (e.g., mice, rats, cats, tree shrews, gray squirrels, and ferrets), as detailed in Table 3. We first find that V1 RFs density ($\rho_{RF}$) acts as a linear classifier ($y = 4.42 \times 10^4 x$), effectively distinguishing species with pinwheel structures from those with salt-and-pepper organizations. In this classifier, species like macaques, cats, tree shrews, and ferrets, which have higher RFs density, are associated with pinwheel structures (light red area in Fig. 5) and exceed the classification threshold. In contrast, species with lower RFs density, such as mice, rats, and gray squirrels, are linked to salt-and-pepper organizations (light blue area in Fig. 5). Thus, V1 RFs density serves as a predictive metric for V1 organizational patterns across species. The $\rho_{RF}$ is calculated as follows:

$$\rho_{RF} = \frac{n'}{s'_r} \propto \frac{n}{\left[ (s_{RF} - \varepsilon)(\sqrt{n} - 1) + s_{RF} \right]^2}, \tag{15}$$

where $n'$ denotes the total number of neurons in V1, $s'_r$ indicates the retinal surface area. We have $n' = s_{V1} \times \rho_{V1}$, where $s_{V1}$ corresponds to the V1 2D surface area, and $\rho_{V1}$ signifies the neuronal density within V1. The variable $\varepsilon$ quantifies the degree of visual input overlap among adjacent neurons, $n$ denotes the total number of neurons, and $s_{RF}$ represents the RF size in the self-evolving spiking neural network (SESNN).

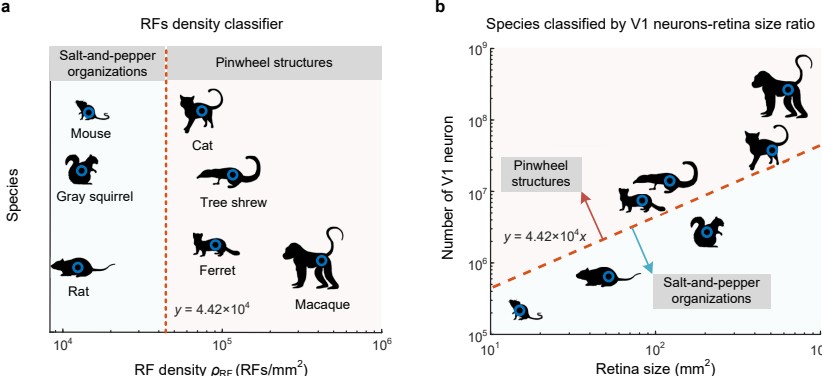

Figure 5: A linear classifier based on RFs density ($y = 4.42 \times 10^4 x$) effectively differentiates species with salt-and-pepper organizations (rats, mice, gray squirrels) from those with pinwheel structures (macaques, ferrets, cats, tree shrews). **a.** This classifier reflects variations in V1 organizations across species. **b**. A plot categorizing species by the ratio of V1 neuron number to retina size acts as a divider, implying a critical ratio for the formation of pinwheel structures.

### A.3.2 SESNN REVEALS THE INFLUENCE OF NEURONAL CONNECTION RANGE ON V1 ORGANIZATIONS

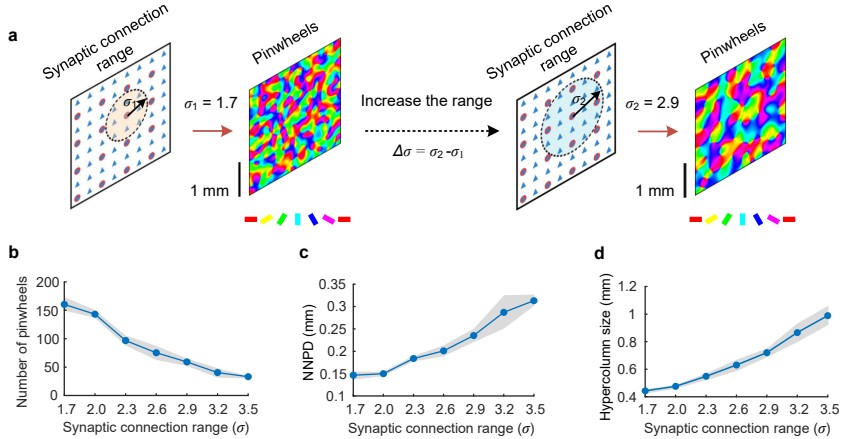

Figure 6: Synaptic connection range within V1 contributes to the formation of pinwheel structures. **a**. Modifying the synaptic connection range reshapes the dimensions of pinwheel structures. **b-d**. The relationship between the synaptic connection range ($\sigma$) and the number of pinwheels, nearest-neighbor pinwheel distance (NNPD) (mm), and hypercolumn size (mm). The scale bar: 1 mm in V1 cortical surface. Color scheme: orientation preference. Lines: mean. Shaded area: SD.

The anatomical data in Table 3d for seven species show variability in V1 neuronal density ($\rho_{V1}$), which influences inter-neuronal spacing and connection strength. We explore how V1 cortical orientation patterns form by adjusting the lateral connection range, impacting axon reach among E- and I-neurons, as depicted in Fig. 6. We modulate axonal arborization through parameter $\sigma$ to adjust the connection range, allowing us to simulate neuronal connections in areas with varying densities. This setup enables the SESNN model to predict changes in cortical patterns (Fig. 6). Our observations indicate that increasing axon lengths, thereby extending the connection range, enlarges hypercolumn sizes within pinwheel structures (Fig. 6d), reduces the overall number of pinwheels (Fig. 6b), and increases NNPD (Fig. 6c). These findings underscore the critical role of neural synaptic connection range in organizing orientation maps.

A.4    QUANTITATIVE DATA ON CORTICAL PINWHEEL STRUCTURES

The visual input overlap is quantified as follows:

$$\varepsilon'_{\text{ratio}} = \frac{\varepsilon'}{s'_{\text{RF}}} = \frac{\sqrt{\rho_{\text{V1}} S_{\text{unit}}} - \frac{L_{\text{unit}} M^{-1}}{s'_{\text{RF}}}}{\sqrt{\rho_{\text{V1}} S_{\text{unit}}} - 1} \tag{16}$$

where $S_{\text{unit}}$ represents the unit cortical area (mm$^2$), $s'_{\text{RF}}$ denotes the size of the RF in V1 with unit in degrees, $\varepsilon'$ denotes the visual overlap degree (in degrees) between neighboring neurons, and $\rho_{\text{V1}}$ represents the density of neurons in V1. We consider only an effective cortical layer composed of output neurons. This is because the apparent overlap within a vertical cortical column primarily contributes to intermediary processing stages for the same input. Therefore, such overlaps should not be conflated with overlaps in the input space. $M$ refers to the CMF. The trend shown in Fig. 2e indicates increasing overlap for mice, macaques, and cats, which is consistent with the overlap settings of our model. To express the visual input overlap ratio as a percentage, we multiply the ratio by 100%. The escalation in overlap corresponds to the expansion of iso-orientation domains and pinwheels, presented by our model's predictions and anatomically detailed by Najafian et al.(Najafian et al., 2022).

A.5    STABILITY AND ADAPTABILITY OF PINWHEEL STRUCTURES

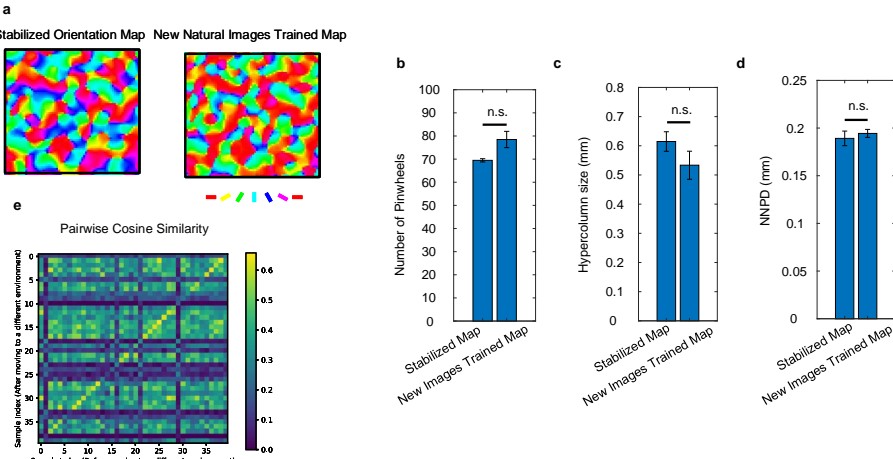

Figure 7: Plasticity of the pinwheel structure in response to new natural images. (a) Stabilized orientation map and orientation map after training with new natural images. (b) Number of pinwheels. (c) Hypercolumn size. (d) NNPD. (e) Pairwise cosine similarity between neuronal representations before and after introducing new environments.

We conducted additional experiments to examine the plasticity of the pinwheel structure in response to new visual environments. A new set of natural images, including forests, large stones, tree branches and leaves, reeds, stones of various sizes, tree trunks, leaves, and grasses, was introduced after the pinwheel structure had stabilized. After further training with these new images, we observed slight modifications in some finer aspects of the pinwheel structure; however, the overall structure, including the locations of the pinwheel centers, remained largely stable (Fig. 7a).

To quantitatively assess these changes, we employed three different metrics: (i) the number of pinwheels, (ii) hypercolumn size, and (iii) NNPD. These metrics were used to compare the pinwheel structure before and after introducing the new images (Fig. 7b, c, d). Additionally, we utilized pairwise cosine similarity to measure the representation similarity of the neuronal spike train population (Fig. 7e).

Although we did not observe significant changes in the morphology of the pinwheels, some detailed aspects of the pinwheel structure did change. Specifically, while the locations of the pinwheel centers

and the overall domains remained nearly unchanged, some neurons exhibited tuning adjustments towards specific orientations. Furthermore, the representation of neurons to the same stimulus remained overall stable, with minor variations before and after the introduction of new environments, suggesting that the pinwheel structure is highly adaptable (Fig. 7).

These results indicate that the model's organizational structure is deeply rooted in the statistical features of natural images and demonstrates a strong capacity for adaptation to new environments while preserving its core structural properties.

### A.6 ORIENTATION MAP FORMATION FROM OUR PROPOSED SPIKING NETWORK MODEL

#### A.6.1 HO RULE LEARNS V1 RF

The HO rule learns RFs from visual images, as described by the following formula:

$$\Delta FF_{im}^{(\text{E}\leftarrow\text{Image})} \propto y_i x_m - y_i^2 FF_{im}^{(\text{E}\leftarrow\text{Image})}, \tag{17}$$

where $x_m$ and $y_i$ represent the gray value of pixel $m$ and instantaneous firing rate of post-synaptic neuron $i$. This rule ensures that weights converge when:

$$FF_{im} = \frac{\langle y_i x_m \rangle}{\langle y_i^2 \rangle} = \frac{\text{STA}_i}{\langle y_i^2 \rangle}, \tag{18}$$

where the operator $\langle \cdot \rangle$ denotes the moving average value (refer to Eq. 4), which corresponds to the normalized spike-triggered average (STA). The HO rule stabilizes learning by constraining synaptic growth, resulting in biologically realistic V1-like RFs.

#### A.6.2 RETINOTOPIC TOPOGRAPHIC CONSTRAINTS

Each neuron encodes only a small portion of the visual field (Fig. 1a), with the small patch $\in \mathbb{R}^{H'\times W'}$ smaller than the input image $I \in \mathbb{R}^{H\times W}$. To represent the entire visual stimulus, the population of neurons collectively tiles the input space through their overlapping visual fields. This organization mirrors the retinotopic mapping observed biologically, where neighboring neurons in the cortex encode adjacent regions of the visual scene, preserving spatial relationships from the retina.

#### A.6.3 HIGH-OVERLAP STRENGTHENS LOCAL E-E CONNECTIONS FOR ORIENTATION MAP CONTINUITY

E-E connections align adjacent neurons' firing activities, promoting similar orientation preferences. This requires a high overlap (Fig. 2), where correlated feedforward input initially strengthens local excitatory connections, ensuring continuous orientation preference and forming iso-orientation domains with similar orientation preferences (Fig. 3e-h).

#### A.6.4 CM RULE VIA E-I-E LOOPS ENSURES ORIENTATION DIVERSITY AND MAP COVERAGE

E-I, I-E, and I-I connections are updated through the CM rule:

$$\Delta W_{ij}^{(\text{K}\leftarrow\text{K}^*)} \propto y_i x_j - \langle y_i \rangle \langle x_j \rangle \left(1 + W_{ij}^{(\text{K}\leftarrow\text{K}^*)}\right), \tag{19}$$

where $x_j$ and $y_i$ represent the instantaneous firing rates of pre- and post-synaptic neuron $j$ and $i$.

At equilibrium, the CM rule converges to:

$$W_{ij} = \frac{\langle y_i x_j \rangle - \langle y_i \rangle \langle x_j \rangle}{\langle y_i \rangle \langle x_j \rangle}. \tag{20}$$

We have:

$$W_{ij} \propto \langle y_i x_j \rangle - \langle y_i \rangle \langle x_j \rangle = \text{Cov}(y_i, x_j). \tag{21}$$

Covariance shapes neural connectivity based on activity covariance, where synaptic weights are strengthened between correlated neurons and weakened between uncorrelated ones. This mechanism

ensures that inhibitory neurons regulate overall activity between neighboring E-neurons through the E-I-E loops (lateral inhibition), suppressing predictable responses to maintain sparse coding. This also prevents excitatory neurons from excitatory connections overly responding to the same orientation, enabling preference for other orientations and ensuring full coverage. Over larger connection ranges, highly correlated feedforward correlations are disrupted and inhibitory neurons drive diverse responses, resulting in distinct domains with varying preferred orientations, forming pinwheel structures within the functional orientation map.

The exact size of the domain, as explained earlier, is determined by the extent of the E-connections; while their swirling patterns are instead demanded by the rich local orientations in each stimulus such that the animal would be able to perceive different orientations in the same local visual field but still satisfying most of the continuity locally. Note that in salt-and-pepper organizations, the mechanisms are disrupted due to reduced overlap. The local continuity of iso-orientation domains is consistent with the HO rule and the overlap between RFs. The domain size is determined by the extent of E-connections, while swirling patterns arise from the diverse local orientations in stimuli. This allows animals to perceive varying orientations within the same local visual field while maintaining local continuity. In salt-and-pepper organizations, reduced overlap disrupts these mechanisms. Hence, visual stimulus overlap critically shapes orientation preference maps in V1: greater overlap promotes pinwheel structures, while reduced overlap leads to salt-and-pepper patterns.

## A.7 INTEGRATE INTO MODERN DEEP LEARNING ALGORITHMS

To demonstrate the practical relevance of our model, we explore its capability as a front-end encoder for deep spiking neural networks. Specifically, we design a five-layer convolutional and pooling-based SNN to classify images from the FashionMNIST dataset. The classification accuracy achieved using our model's spike sequences is compared with that of Poisson spike sequences, as summarized in Table 4.

Table 4: Pinwheels vs. Poisson encoder accuracy (FashionMNIST).

| Dataset class | Pinwheels encoder accuracy(%) | Poisson encoder accuracy(%) |
|---|---|---|
| T-shirt / top | 86.80 | 85.40 |
| Trouser | 97.30 | 96.40 |
| Pullover | 83.40 | 83.80 |
| Dress | 91.10 | 90.30 |
| Coat | 83.60 | 81.60 |
| Sandal | 96.20 | 96.80 |
| Shirt | 73.20 | 67.80 |
| Sneaker | 95.40 | 95.40 |
| Bag | 97.80 | 97.00 |
| Ankle boot | 96.10 | 94.20 |
| Overall | 90.09 | 88.87 |

From these results, we observe that, compared to the randomness inherent in Poisson-based encoding, the functional organization of pinwheels produces spike sequences with more structured spatiotemporal patterns. This organization encodes richer information, contributing to improved classification accuracy across most categories.

Additionally, we evaluate the noise robustness of our model by introducing Gaussian noise to the input images. The results are shown in Table 5.

Compared to the Poisson encoder, the functional organization inherent in pinwheel encoding provides better resistance to noise, as it retains meaningful features even in noisy environments. Moreover, the salt-and-pepper encoding also shows a degree of robustness, albeit slightly less effective than pinwheel encoding under higher noise levels.

These findings highlight that the spatiotemporal regularity of the pinwheel maps significantly benefits both classification performance and noise resistance, suggesting their potential utility in enhancing the robustness of modern deep learning frameworks.

Table 5: Noise robustness: Pinwheels vs. Poisson vs. Salt-and-peppers encoders.

| Noise level | Pinwheels encoder accuracy (%) | Poisson encoder accuracy (%) | Salt-and-peppers encoder accuracy (%) |
|---|---|---|---|
| 0.0 | 89.47 | 93.14 | 87.92 |
| 0.1 | 88.98 | 92.14 | 87.21 |
| 0.2 | 88.41 | 82.81 | 86.18 |
| 0.3 | 88.11 | 62.79 | 84.51 |
| 0.4 | 86.22 | 36.61 | 80.52 |
| 0.5 | 82.91 | 18.79 | 67.39 |
| 0.6 | 70.88 | 12.03 | 38.37 |
| 0.7 | 43.60 | 10.89 | 19.95 |
| 0.8 | 21.20 | 10.50 | 13.00 |
| 0.9 | 12.51 | 10.20 | 10.71 |

## A.8 COMPUTING INFRASTRUCTURE

Simulations are executed on a high-performance system featuring an Intel® Xeon® Gold 6348 CPU (2.60 GHz), an NVIDIA® A100 GPU, and 512 GB of memory. The workflow was managed on Ubuntu 20.04.6 LTS, with computational tasks implemented in MATLAB R2023a and Python 3.9.

Table 6: Computing infrastructure.

| | |
|---|---|
| CPU | Intel® Xeon(R) Gold 6348 CPU @ 2.60GHz |
| GPU | A100 |
| Memory | 512 GB |
| Operating system | Ubuntu 20.04.6 LTS |
| Simulation platform | MATLAB R2023a and Python 3.9 |

## A.9 ACRONYM

Table 7 summarizes essential acronyms and their full terms, spanning neural structures (e.g., V1), computational models (SESNN, SNN), learning rules (HO, CM), and analytical metrics (NNPD, ICE, SI).

Table 7: List of acronyms and their corresponding terms.

| Acronym | Term |
|---|---|
| V1 | Primary Visual Cortex |
| SESNN | Self-Evolving Spiking Neural Network |
| SNN | Spiking Neural Network |
| HO | Hebbian Oja's variant |
| CM | Correlation Measuring |
| NNPD | Nearest-neighbor pinwheel distance |
| ICE | Information-Cost Efficiency |
| CMF | Cortical Magnification Factor |
| E-I | Excitatory-Inhibitory |
| FF | Feedforward |
| RF | Receptive Field |
| SI | Stability index |

