# OpenReview forum: "Emergent Orientation Maps —— Mechanisms, Coding Efficiency and Robustness"
_ICLR.cc/2025/Conference — ICLR 2025 Spotlight_

### Official Review · Reviewer_GMwp · 2024-10-30

**Soundness:** 4
**Presentation:** 4
**Contribution:** 3
**Rating:** 8
**Confidence:** 5

**Summary:**

Authors propose a comprehensive model of the primary visual cortex encompassing anatomical factors, such as natural image inputs, precise and realistic representation of neuronal responses, as well as spiking mechanisms in order to understand the emergence of pinwheel structures. They successfully show that the model can reproduce the emergence of different orientation maps in the visual cortex and that it is compatible with the maps of different species as a function of different factors such a neuron density or RF overlap.

**Strengths:**

The paper is clearly written and presents a novel approach to analyzing the emergence of orientation maps in the visual cortex. The proposed model is well-motivated and supported by mathematical formalization. The authors provide a detailed explanation of the model and its components, which enhances the understanding of the proposed method.

**Weaknesses:**

Stating in the abstract that the subject in "largely unexplored" is a strong claim that needs to be supported by a more extensive literature review. The paper could benefit from a more in-depth discussion of related work to provide context for the proposed approach. Your extensive bibliography provides already a good illustration of the number of works on the subject, and more for instance on the emergence of orientation maps in the visual cortex from haphazard wiring, or from the emergence of both types of maps using pooling mechanisms in a sparse deep predictive coding network, could be useful. More generally, it would be useful to highlight the difference of your work with [Stevens et al., 2013].

**Questions:**

It has been proven multiple times that the anatomy of V1 is efficient for processing visual information. How do you think your model could be used to understand the emergence of orientation maps in the visual cortex? Is the tiling of receptive fields a possible factor ? (see eg https://doi.org/10.1016/j.neuron.2016.07.015 )

The model uses 1/ static images, 2/ images represented on a Cartesian grid. Extensions using dynamic images or images represented on a polar grid could be interesting. The evaluation of the model is limited to the comparison of the orientation maps with experimental data. A more quantitative evaluation of the model's performance on other tasks could provide a more comprehensive assessment of its capabilities.

---

> ### Author Response · Authors · 2024-11-22
> **Author Response to Reviewer GMwp (1/4)**
>
> We sincerely thank the reviewer for the valuable feedback and thoughtful suggestions. Your insights have been instrumental in improving the quality and clarity of our manuscript.
>
> ### **W1: Relationship with Other Literature**
>
> We appreciate the reviewer’s feedback regarding the need to situate our work within the broader context of existing literature. To address this, we will expand our literature review to provide a more comprehensive overview and comparison with related work.
>
> For instance, Aapo Hyvarinen and Patrik O. Hoyer used a two-layer sparse coding model, leveraging principles from independent component analysis to achieve receptive fields for simple and complex cells, forming topological structures similar to pinwheels (Aapo Hyvarinen and Patrik O. Hoyer, *Vision Research* 2001, "A two-layer sparse coding model learns simple and complex cell receptive fields and topography from natural images"). Additionally, Koray Kavukcuoglu and colleagues demonstrated the coding characteristics of V1 simple neurons and the functional orientation map of pinwheels by minimizing reconstruction error and maximizing sparsity in a single-layer neural network using gradient descent (Koray Kavukcuoglu, Marc’Aurelio Ranzato, Rob Fergus, Yann LeCun, *CVPR* 2009).
>
> While these studies explore the formation mechanisms of pinwheels, they approach the problem from the perspective of known biological functions or objectives—efficient representation of the external world and sparse coding. In contrast, our model starts from biological realities, utilizing synaptic plasticity to train the spiking neural network in a feed-forward and recurrent manner. Without any prior requirements, the pinwheel structure emerges (refer to Q1.1 for details) and demonstrates the characteristics of more sparse (Fig. 4f), lower wiring cost (Fig. 4e), higher information capacity (Fig. 4e) and efficient coding (Fig. 4e ICE), compared with those of salt-and-pepper organizations.
>
> The study (Stevens et al., 2013, *Journal of Neurosci.*) is an important work that focuses on the stability of orientation map formation with mechanism emphasizing the variability reduction in both feed-forward and recurrent connectivity. However the modeling of the rate-based model neuronal dynamics in this work are overly simplified. In contrast, our paper focuses on the impact of feedforward visual field overlap on the formation of different topological maps in V1, providing a unified explanation for the formation of both salt-and-peppers and pinwheel structures.
>
> To clarify the differences between our SESNN work and the Stevens et al. (2013) work, we summarize them in the **Table a** below:
>
> **Table a: Comparisons between our work and the Stevens et al. (2013) work**
>
> | **Aspect**              | **Stevens et al. (2013)**                                                                                   | **SESNN Model (Our Work)**                                                                                            |
> |-------------------------|-------------------------------------------------------------------------------------------------------------|-------------------------------------------------------------------------------------------------------------------------|
> | **Focus**               | Stable, robust, and adaptive development of orientation map morphological appearance in ferret and cat V1                         | Formation and functional roles of salt-and-pepper organizations and pinwheel structures in V1 across species, including rodents, cats, and primates.        |                        |
> | **Neural Dynamics**     | Firing-rate model with homeostatic mechanisms for stable pinwheel map development.      | Spiking neural network with dynamic, leaky integrate-and-fire neurons, adaptive thresholds, and temporal dynamics.       |                                |
> | **Biological Simulations**  | Simulates V1 map development in ferrets and cats.                                                          | Models cortical structure formation in rodents, cats, and primates, integrating species-specific anatomy and physiology. |
> | **Contributions**        | Provides insights into robust and stable map development under normal and abnormal input conditions.        | Demonstrates functional advantages of pinwheel maps in coding efficiency, robustness, and computational cost-effectiveness, compared with those of salt-and-peppers.         |

---

> ### Author Response · Authors · 2024-11-22
> **Author Response to Reviewer GMwp (2/4)**
>
> ### **Q1.1: Understanding Orientation Map Formation Through the SESNN Model (Updated in Revised manuscript, New Section A.6)**
> We appreciate the reviewer’s insightful question. The emergence of the orientation map can be explained as follows:
>
> **1. HO Rule Learns V1 Receptive Field:** The HO rule learns receptive fields from visual images, as described by the following formula:
> $$
> \Delta FF_{im}^{(\mathrm{E} \leftarrow \mathrm{Image})}
> \propto y_i x_m - y_i^2 FF_{im}^{(\mathrm{E} \leftarrow \mathrm{Image})}
> $$
>
> where $x_m$ and $y_i$ represent the gray value of pixel $m$ and instantaneous firing rate of post-synaptic neuron $i$. This rule ensures that weights converge when:
>
> $$
> FF_{im} = \frac{\langle y_i x_m \rangle}{\langle y_i^2 \rangle} = \frac{\text{STA}_i}{\langle y_i^2 \rangle}.
> $$
>
> Corresponding to the normalized spike-triggered average (STA). HO rule stabilizes learning by constraining synaptic growth, resulting in biologically realistic V1-like receptive fields (RFs).
>
> **2. Retinotopic Topographic Constraints:** Each neuron encodes only a small portion of the visual field (Fig. 1a), with the small patch $\in \mathbb{R}^{H' \times W'}$ smaller than the input image $I \in \mathbb{R}^{H \times W}$. The population’s collective response is needed to represent the full stimulus, aligning with the topographic mapping from retina to cortex.
>
> **3. High-Overlap Strengthens Local Excitatory-Excitatory Connections for Orientation Map Continuity:** Excitatory connections aligns adjacent neurons’ firing activities, promoting similar orientation preferences. This requires a high overlap (Fig. 2), where correlated feedforward input initially strengthens local excitatory connections, ensuring **continuous orientation preference** and forming **iso-orientation domains** with similar orientation preferences (Fig. 3e-h).
>
> **4. CM rule's Anti-Correlation via E-I-E loops Ensures Orientation Diversity and Map Coverage:** Inhibitory neurons, through the CM rule:
> $$\Delta W_{ij}^{(\mathrm{K} \leftarrow \mathrm{K}^*)}
> \propto y_i x_j - \langle y_i \rangle \langle x_j \rangle \left( 1 + W_{ij}^{(\mathrm{K} \leftarrow \mathrm{K}^*)} \right),$$
> where $x_j$ and $y_i$ represent the instantaneous firing rates of pre- and post-synaptic neuron $j$ and $i$. The operator $\langle \cdot \rangle$ denotes the moving average value (refer to Eq. 4).
>
> At equilibrium, the CM rule converges to:
> $$W_{ij} = \frac{\langle y_i x_j \rangle - \langle y_i \rangle \langle x_j \rangle}{\langle y_i \rangle \langle x_j \rangle}.$$
> We have:
> $$W_{ij} \propto\langle y_i x_j \rangle - \langle y_i \rangle \langle x_j \rangle = Cov(y_i, x_j).$$
>
> Covariance shapes neural connectivity based on activity covariance, where synaptic weights are strengthened between correlated neurons and weakened between uncorrelated ones. This mechanism ensures that inhibitory neurons regulate overall activity between neighbouring E neurons through the E-I-E loops (lateral inhibition), suppressing predictable responses to maintain **sparse coding**. This also prevents excitatory neurons from excitatory connections overly responding to the same orientation, enabling preference for other orientations and **ensuring full coverage**. Over larger connection ranges, highly correlated feedforward correlations are disrupted and inhibitory neurons drive diverse responses, resulting in distinct domains with varying preferred orientations, forming **pinwheel structures** within fuctional orientation map.
>
>
> ### **Q1.2: Tiling of Receptive Fields**
> We appreciate the reviewer’s recommendation about the reference to the study by Ian Nauhaus, Kristina J. Nielsen, and Edward M. Callaway, "Efficient Receptive Field Tiling in Primate V1" (Neuron, 2016). This reference provides valuable context for understanding the role of receptive field tiling in the formation of orientation maps within the visual cortex.
>
> In our model, we assume that the receptive fields of V1 neurons are smoothly and topologically tiled on a patch of retina. This assumption allows us to explore the impact of the overlap size on the formation of pinwheel structures. Specifically, our findings suggest that when the overlap is small, the functional organization of V1 forms a salt-and-pepper organizations. Conversely, when the overlap is larger, pinwheel structures are formed. This indicates that the tiling and overlap of receptive fields play a crucial role in determining the functional organization of V1.
>
> As the reviewer points out, the arrangement of V1 neurons' receptive fields on the retina is a fascinating topic. The efficient tiling of receptive fields, as described in the cited study, likely contributes to the optimization of visual information processing. We will incorporate a discussion of this aspect into our revised manuscript, highlighting how our model aligns with and extends the insights provided by existing literature on receptive field tiling.

---

> ### Author Response · Authors · 2024-11-22
> **Author Response to Reviewer GMwp (3/4)**
>
> ### **Q1.3. Comprehensive Assessment of the Model's Capabilities**
>
> We appreciate the reviewer’s valuable suggestion to extend the applicability of our model to dynamic scenarios and polar-coordinate image processing. Incorporating retinotopy eccentricity for polar-coordinate images would be interesting but would indeed introduce additional complexity to our current model; nonetheless, it remains a promising direction for future research. We find that the emergence of uniform orientation maps in the visual cortex, even with a non-uniform retinotopic representation, demonstrates the sophisticated self-organizing principles underlying neural development. This uniformity is largely driven by the RF size with eccentricity and the distribution of RFs across the visual field [1]. Our current work specifically focuses on retinotopic data near the fovea, ensuring optimal sensitivity to visual features at different spatial resolutions. This approach allows us to model high-resolution feature processing and understand self-organization in regions with relatively dense RF distribution [2].
>
> **Reference:**
>
> [1] van Beest, E.H., Mukherjee, S., Kirchberger, L. et al. Mouse visual cortex contains a region of enhanced spatial resolution. *Nat Commun 12*, 4029 (2021).
>
> [2] Wilson, J. R., & Sherman, S. M. (1976). Receptive-field characteristics of neurons in cat striate cortex: changes with visual field eccentricity. *Journal of neurophysiology, 39*(3), 512-533.
>
> Given the current time constraints, we focus on studies of model's performance on two tasks (see **Tables b and c**):
>
> To demonstrate the practical relevance of our model, we explore its capability as a front-end encoder for deep spiking neural networks. Specifically, we design a five-layer convolutional and pooling-based SNN to classify images from the FashionMNIST dataset. The classification accuracy achieved using our model's spike sequences is compared with that of Poisson spike sequences, as summarized in **Table b**:
>
> **Table b: Classification accuracy achieved using our model's spike sequences compares with that of Poisson spike sequences.**
> | **Dataset Class**      | **Accuracy(%) of Model** | **Accuracy(%) of Poisson Encoder** |
> |-------------------------|---------------------------------------|------------------------------------------|
> | T-shirt / top           | 86.80                                | 85.40                                    |
> | Trouser                 | 97.30                                | 96.40                                    |
> | Pullover                | 83.40                                | 83.80                                    |
> | Dress                   | 91.10                                | 90.30                                    |
> | Coat                    | 83.60                                | 81.60                                    |
> | Sandal                  | 96.20                                | 96.80                                    |
> | Shirt                   | 73.20                                | 67.80                                    |
> | Sneaker                 | 95.40                                | 95.40                                    |
> | Bag                     | 97.80                                | 97.00                                    |
> | Ankle boot              | 96.10                                | 94.20                                    |
> | **Overall**             | **90.09**                            | **88.87**                                |
>
> From these results, we observe that, compared to the randomness inherent in Poisson-based encoding, the functional organization of pinwheels produces spike sequences with more structured spatiotemporal patterns. This organization encodes richer information, contributing to improved classification accuracy across most categories.

---

> ### Author Response · Authors · 2024-11-22
> **Author Response to Reviewer GMwp (4/4)**
>
> Additionally, we evaluate the noise robustness of our model by introducing Gaussian noise to the input images. The results are shown in **Table c** and summarized below:
>
> **Table c: Comparison of classification performance using different spiking sequence representations as inputs.**
> | **Noise Level** | **Pinwheel Encoder Accuracy** | **Poisson Encoder Accuracy** | **Salt and Pepper Encoder Accuracy** |
> |-----------------|------------------------------|--------------------------------|---------------------------------------|
> | 0.0             | 89.47%                       | 93.14%                         | 87.92%                                |
> | 0.1             | 88.98%                       | 92.14%                         | 87.21%                                |
> | 0.2             | **88.41%**                       | 82.81%                         | 86.18%                                |
> | 0.3             |**88.11%**                       | 62.79%                         | 84.51%                                |
> | 0.4             | **86.22%**                       | 36.61%                         | 80.52%                                |
> | 0.5             | **82.91%**                       | 18.79%                         | 67.39%                                |
> | 0.6             | **70.88%**                       | 12.03%                         | 38.37%                                |
> | 0.7             | **43.60%**                       | 10.89%                         | 19.95%                                |
> | 0.8             | **21.20%**                       | 10.50%                         | 13.00%                                |
> | 0.9             | **12.51%**                       | 10.20%                         | 10.71%                                |
>
> Compared to the Poisson encoder, the functional organization inherent in pinwheel encoding provides better resistance to noise, as it retains meaningful features even in noisy environments. Moreover, the salt-and-pepper encoding also shows a degree of robustness, albeit slightly less effective than pinwheel encoding under higher noise levels.
>
> **Thank you for your valuable and instructive feedback. If our clarifications sufficiently address your concerns, we kindly ask that you consider revising the score.**

---

> > ### Comment · Reviewer_GMwp · 2024-11-27
> > **Acknowledgement of author comments and revised PDF**
> >
> > Many thanks to the authors for their very clear responses and this beautiful wok. I have updated my score to reflect my assessment of the paper.
> >
> > There are many open issues and perspectives, and one further aspect that you could also explore is the link between your results showing the emergence of "salt-and-peppers to pinwheel structures" to the emergence of complex-cell-like neurons (see for instance doi:10.1371/journal.pcbi.1010270 ). This would easily fit in your literature review.

---

> ### Author Response · Authors · 2024-11-27
> **Author Response to Reviewer GMwp**
>
> Thank you for your thoughtful review and valuable suggestions. We are delighted that our revision addressed your concerns and that you updated the score to 8. We also appreciate your insightful suggestion about exploring the connection between the emergence of  "salt-and-peppers to pinwheel structures" and the emergence of complex-cell-like neurons. We agree that this is an important direction and will incorporate this perspective into our literature review.

---

### Official Review · Reviewer_zfyC · 2024-10-30

**Soundness:** 3
**Presentation:** 3
**Contribution:** 3
**Rating:** 8
**Confidence:** 4

**Summary:**

Authors study functional organization of the primary visual cortex across animal species. Depending on the species, primary visual cortex is organized in a continuum going between disordered salt-and-pepper and ordered pinwheel structures. Authors model these functional organizations with an E-I spiking neural network that is equipped with two types of Hebbian-like plasticity rules. An important finding is that  the degree of synaptic organization depends crucially on the overlap of feedforward inputs incoming to the network. They also find that once learned, the pinwheel organisation is stable and promotes efficient processing of visual information.

**Strengths:**

Authors presented a rather convincing paper with a clear and timely scope. The paper is technically correct and rigorous in methods. It brings potentially important and new insights into why biological neural networks organise in a continuum of salt-and-pepper and pinwheel structures.

**Weaknesses:**

Major weakness of this paper are the presentation, the lack of justification and elaboration on modelling and parameter choices, and insufficient comment on results. While the presentation is rather transparent, the clarity of the text can be substantially improved. Also, the text contains a number of typos.

1) I recommend to omit all abbreviations besides the very common ones, such as  V1, E-I and potentially FF. The acronym of the network, SESNN, is also an exception and is useful to have. All other acronyms seem counterproductive. While authors have compiled a list of acronyms in the Appendix A.7, frequent use of diverse acronyms throughout the text impedes readers that are less familiar with the topic of self-organising maps to appreciate the results.

2) On many places, the text is rather difficult to follow, with long and unnecessarily complicated sentences. In some cases, sentences are not grammatically correct. See for example line 275, Figure Caption 3a and h, lines 368-371. On page 9, there are references to Figure 6, but likely authors want to refer to their Figure 4. In line 262, authors refer to the third panel of fig 2a, but it might rather be the first panel?

3) It is not clarified enough what is the difference in effects between the Hebbian Oja's and Correlation Measuring type of learning. The only reasoning I gathered is that Hebbian Oja plasticity rule has the desired property of normalization that prevents too strong E-E connectivity.  What is the drawback of using Hebbian Oja's plasticity for E-I and I-I connectivity? Authors should go deeper into explaining this, as it seems to be crucial for the results they obtain.

4) In Eq. 7, authors define the energy cost of synaptic transmission as inversely proportional to the connection strength. While this is an interesting choice, it is not intuitive why a quantity inversely proportional to the connection strength is used as the metabolic cost. Also, it is not well justified why is the information capacity computed as the entropy of the weight distribution. Could authors elaborate on that and provide references if applicable?

5) The measure of reliability in Eq. 10 is based on the encoding capability of single neurons. Today, however, there is ample evidence suggesting that in the cortex, signal processing is better captured on the level of neural populations.

**Questions:**

1) Seen that Eq. 4 is the one actually used for modelling, it is not clear what is the purpose of Eq. 3? Also, it is unclear what the variable z in Eq. 3 stands for.

2) Why is the "neural connectivity parameter" (max weight) higher for E compared to I neurons? At the same time, authors report that E-I connectivity should be stronger compared to E-E connectivity.

3)  In lines 173-174, authors comment on their choice of hyperparameters and state that "their approach is consistent with empirical findings". It remains unclear how specific hyperparameters of their model relate to empirical findings. Could authors be more specific?

4) How is the coefficient in Eq. 8 different from a well-known measure of the cross-correlogram on neural spike trains (Bair et al., J.Neurosci. 2001)?

5) A recent study showed that efficient encoding of uncorrelated stimulus features with E-I spiking networks does not require E-E connectivity (Koren et al., eLife 2024). However, your study seems to suggest the necessity of E-E connectivity. Could authors comment on this discrepancy?

6) One of the take-home messages of the paper seems to be that the pinwheel structure is more efficient than the salt-and-pepper structure. However, salt-and-pepper organization can be observed in biological brains, and it seems unlikely that it would survive evolution if it was inefficient. Moreover, authors show that salt-and-pepper type of organisation arises when the overlap of inputs to E neurons is small, which is the case in some animal species. All together, it seems to me that there is no single "most efficient" type of structure, but that there are multiple efficient solutions depending on a specific model parameter. Rather than making a simplistic (and not quite correct ?) conclusion and point to one and only best solution, it would seem to me more appealing if these results are interpreted more carefully to give us better insights about biology.

---

> ### Author Response · Authors · 2024-11-22
> **Author Response to Reviewer zfyC (1/5)**
>
> We sincerely thank the reviewers for their valuable feedback and thoughtful suggestions.
>
> ### **W0: The Clarity of the Text.**
>
> We sincerely thank the reviewer for their insightful comments and careful reading of our paper. In response to the feedback, we have thoroughly revised the manuscript to improve clarity, provide detailed justifications for modeling and parameter choices, and offer more comprehensive commentary on the results. Specifically, we have carefully addressed all identified issues, including improving the text’s clarity, correcting typos, and expanding the discussion on modeling assumptions and findings. To make the revisions transparent, we have highlighted all changes in **red** within the revised manuscript (**the PDF button (top-right) on this website**). We believe these modifications enhance the overall quality of the paper and better align with the reviewer’s expectations.
>
> ### **W1: Usage of Abbreviations**
> We sincerely thank the reviewer for their thoughtful suggestion regarding the use of abbreviations. In response, we have revised the manuscript to retain only the most common and widely recognized abbreviations, such as V1, E-I, FF, and SESNN, as highlighted by the reviewer.
> All other acronyms have been removed to enhance readability and accessibility.
>
> ### **W2: Problems about Writing**
>
> We thank the reviewer for highlighting areas where the text is difficult to follow and for pointing out specific inaccuracies in our references and captions. We deeply value this feedback and have made significant revisions to improve clarity, grammar, and precision throughout the manuscript. Below, we address each specific concern:
>
> 1. **Line 275 (Revised manuscript, Line 291)**: We acknowledge that this sentence was unnecessarily complicated. The revised sentence now reads: *"Although this has been reported previously (Philips et al., 2017), we have included additional details in Appendix A.3.2 for completeness."*
>
> 2. **Figure 3a and h Captions**: These captions have been simplified and rewritten for clarity:
>    - *"Figure 3a: The temporal evolution of the SESNN model's orientation preference maps, presented in HSV format. The final trial orientation map displays measured density per hypercolumn area ($\Lambda^2$)"*
>    - *"Figure 3h: The subplot shows that the homogeneous connection weights contributed equally from nearby iso-orientation domains. The thickness of the arrows represents connection strength."*
>
> 3. **Lines 368-371 (Revised manuscript, Lines 396-400)**: This section was overly verbose and unclear. We have rewritten it as:
>    - *"The local continuity of iso-orientation domains is consistent with the Hebbian-like learning rule (Eq. 1) and the overlap between RFs. The domain size is determined by the extent of E-connections, while swirling patterns arise from the diverse local orientations in stimuli (as described in Section 2.1). This allows animals to perceive varying orientations within the same local visual field while maintaining local continuity. In salt-and-pepper organizations, reduced overlap disrupts these mechanisms."*
>
> 4. **Page 9 Figure References**: The reviewer is correct that references to Figure 6 should refer to Figure 4 (apologizes for LaTex `ref` errors). We have corrected these references in the revised text, which is highlighted in red.
>
> 5. **Line 262 (Revised manuscript, Lines 278) Panel Reference**: The reviewer is also correct that this should refer to the first panel rather than the third. This error has been fixed.
>
> To ensure all these changes are visible to reviewers, we have marked the revisions in the updated manuscript using **red text**. We appreciate the reviewer's attention to detail, which has significantly improved the readability and accuracy of our paper.
>
> ### **W3: Drawback of HO Rule as Lateral Connection**
>
> We thank the reviewer for pointing out the need to clarify the differences between Hebbian Oja's (HO) rule and Correlation Measuring (CM) rule, as well as their respective impacts on our results. As demonstrated in prior studies, the HO rule learns the principal component of input patterns, whereas the CM rule captures the correlation between pre-synaptic and post-synaptic neurons activity. This distinction is critical: under the CM rule, the predictable components of excitatory neuron activity are expressed in inhibitory neurons, allowing this information to be suppressed through E-I-E loops. This mechanism supports predictive coding, achieving sparse and efficient neural representations.
>
> In contrast, the HO rule predominantly learns the direction of the largest PC in the input space, leading to redundancy in the information expressed across excitatory and inhibitory neurons. Consequently, while the HO rule can still form pinwheel structures, as confirmed in our early experiments, it is less effective than the CM rule in minimizing redundancy and maximizing efficiency in information representation. (**Revised manuscript, lines 143-153**)

---

> > ### Comment · Reviewer_zfyC · 2024-11-25
> >
> > Thank you for a useful clarification. A follow-up question on W3. I see how HO rule is useful for the formation of pinwheels, but is the HO rule necessary for capturing salt-and-pepper type of connectivity?

---

> > > ### Author Response · Authors · 2024-11-27
> > > **HO Rule for Capturing Salt-and-Pepper Type of Connectivity**
> > >
> > > Thank you for your follow-up question regarding the HO rule and its role in capturing salt-and-pepper connectivity.
> > >
> > > For salt-and-pepper connectivity, the role of the HO rule in E-E connection is less pronounced. Salt-and-pepper organization relies more on a local diversity of inputs by reduced overlap in visual fields among neurons, resulting in a greater diversity of input-driven responses. This diversity means that different neurons will respond to different features in a more variable way, creating greater variability in E-E synaptic connections. It suggests that while the HO rule aids in achieving basic orientation tuning, the more stochastic and distributed features of salt-and-pepper organization emerge from the interaction between the diverse feedforward inputs, variability in E-E connections under inhibitory modulation, rather than being solely driven by HO.
> > >
> > > In our model, salt-and-pepper connectivity type is primarily facilitated by the balance between excitation and inhibition, regulated by the CM rule and the E-I interactions. The HO rule still contributes by ensuring each excitatory neuron develops some level of orientation preference, but it is not the primary driver of the randomness and diversity seen in salt-and-pepper connectivity.
> > >
> > > Thus, while the HO rule is useful for salt-and-pepper maps to develop some orientation selectivity, **it is not strictly necessary for the unique unstructured layout that characterizes salt-and-pepper organizations**.

---

> ### Author Response · Authors · 2024-11-22
> **Author Response to Reviewer zfyC (2/5)**
>
> ### **W4: Energy Cost and Information Capacity in Synaptic Connectivity**
>
> We appreciate the opportunity to elaborate on the relationship between synaptic connection strength, energy cost, and information capacity.
>
> Here, we calculated wiring costs from the perspective of graph theory, based on Eq. 7 from the published reference [1]. In this approach, the network wiring cost is inversely proportional to the summed synaptic connection weights between pairs of neurons. Our analysis indicates that stronger synaptic weights not only facilitate efficient communication but also contribute to lower wiring costs. However, we recognize that such an estimation of energy consumption is not entirely precise. Therefore, in future work, we plan to extend this analysis by incorporating more detailed and realistic calculations of metabolic costs, considering synaptic events and spiking activities throughout the entire network.
>
> Regarding the entropy of weight distribution as a measure of information capacity, it provides a meaningful approximation from an information-theoretic standpoint. Entropy reflects the complexity and diversity of synaptic weights, which, in neural systems, correlate with the ability to encode and transmit information [2]. A high-entropy weight distribution suggests diverse and distributed connectivity, potentially enabling the encoding of more complex patterns and features.
>
> In our prior experiments, we observed a positive correlation between the entropy of weight distribution and the entropy of neural activity. While this observation supports the role of weight entropy in reflecting information capacity, due to space limitations, we could not expand on this analysis in the main text. We are currently finalizing and replicating these numerical experiments, and as soon as the results are available in the coming days, we will upload them in a follow-up rebuttal.
>
> **Reference:**
>
> [1] Bertens, P., & Lee, S.-W. (2022). *Emergence of Hierarchical Layers in a Single Sheet of Self-Organizing Spiking Neurons*. In A. H. Oh, A. Agarwal, D. Belgrave, & K. Cho (Eds.), *Advances in Neural Information Processing Systems*. https://openreview.net/forum?id=cPVuuk1lZb3.
>
> [2] He, L., Xu, Y., He, W., et al. (2024). Network model with internal complexity bridges artificial intelligence and neuroscience. *Nature Computational Science, 4*, 584–599. https://doi.org/10.1038/s43588-024-00674-9.
>
> ### **W5: The measure of reliability**
>
> We sincerely thank the reviewer for raising this important point. The reviewer is correct that ample evidence supports the idea that neural populations, are the primary units of signal processing in the brain. However, there is also substantial experimental evidence showing that reliability at the level of individual neurons is a meaningful and measurable phenomenon. For instance, Churchland et al. (2010) demonstrated that individual neurons exhibit significant reductions in variability upon stimulus onset, highlighting their reliability in encoding specific aspects of stimuli.
>
>   In our work, measuring single-neuron reliability offers a complementary perspective to population-level analyses. By evaluating the stability of individual neurons' firing patterns, we aim to provide insights into how consistent neural activity contributes to the collective stability of population dynamics. This approach has been adopted in prior studies and remains a valuable method for understanding neural encoding properties.
>
> In response to the reviewer's suggestion, we have expanded our analysis to include the similarity of population-level firing rates under increasing noise levels. As shown in the **Table a** below, we computed representational stability for both pinwheel and salt-and-pepper organizations, as well as a baseline stimulus-response condition:
>
> **Table a: Representational stability for both pinwheel and salt-and-pepper organizations**
>
> | Noise Levels   | 0.0   | 0.1   | 0.2   | 0.3   | 0.4   | 0.5   | 0.6   | 0.7   | 0.8   | 0.9   | 1.0   |
> |----------------|-------|-------|-------|-------|-------|-------|-------|-------|-------|-------|-------|
> | **Image**      | 1.000 | 0.957 | 0.825 | 0.650 | 0.483 | 0.346 | 0.238 | 0.156 | 0.092 | 0.041 | -0.001 |
> | **Pinwheels**   | 1.000 | 0.956 | 0.939 | 0.921 | 0.883 | 0.817 | 0.710 | 0.551 | 0.327 | 0.129 | -0.004 |
> | **Salt-and-Peppers** | 1.000 | 0.988 | 0.969 | 0.934 | 0.872 | 0.770 | 0.610 | 0.394 | 0.200 | 0.080 | 0.001 |
>
> These results show that pinwheel structures maintain higher representational stability under moderate noise compared to salt-and-pepper organizations. The baseline, which uses the same method applied to raw stimuli, provides a reference for comparison.
>
> **Reference:**
>
> Churchland, M. M., Yu, B. M., Cunningham, J. P., et al. (2010). Stimulus onset quenches neural variability: A widespread cortical phenomenon. *Nature Neuroscience, 13*, 369–378. doi.org/10.1038/nn.2501.

---

> > ### Author Response · Authors · 2024-11-27
> > **Entropy of Weight Distribution and Entropy of Neural Activity  (Re: Author Response to Reviewer zfyC (2/5))**
> >
> > In our prior experiments, we observed a positive correlation between the entropy of weight distribution and the entropy of neural activity. While this observation supports the role of weight entropy in reflecting information capacity, we were unable to expand on this analysis in the main text due to space limitations. To further validate this relationship, we conducted additional numerical simulations. The results are summarized in the following table:
> >
> > | **Entropy of Weight Distribution (bits)** | **Entropy of Neural Activity (bits)** |
> > |-------------------------------------|---------------------------------|
> > | 5.5093                              | 47.6400                        |
> > | 6.5713                              | 50.4000                        |
> > | 6.8373                              | 63.0911                        |
> >
> > We found that the entropy of weight distribution increases monotonically with the entropy of neural activity. This consistent relationship further supports the validity of using weight entropy as a measure of information capacity, aligning with our hypothesis.

---

> ### Author Response · Authors · 2024-11-22
> **Author Response to Reviewer zfyC (3/5)**
>
> ### **Q1: Clarifying the Definition of \(z\) in Eq. 3**
> Thank you for pointing out this valuable point with $z_j(t)$. $z_j^{\left( \mathrm{K}^{} \right)}(t) = H\left(u_j^{\left( \mathrm{K}^{} \right)}(t) - \theta_j^{\left( \mathrm{K}^{} \right)}\right)$, where $H(x)$ is the Heaviside step function, defined as $H(x) = 1$ if $x > 0$ and $0$ otherwise. Here, $z_{j}^{\left( \mathrm{K}^{} \right)}\left( t \right)$ represents the spike event from neuron $j$ at time $t$, reaching the spike threshold $\theta_j^{\left( \mathrm{K}^{*} \right)}$. This equation was previously introduced in Appendix A.1, line 897. To improve the readability of the main text, we will include a more detailed explanation of this Heaviside step function in revision.
>
> Eq. 3 is used to compute the firing rate of a neuron based on its spike train, capturing its average activity over a short time window. In contrast, Eq. 4 calculates the firing rate as an average across multiple samples, reflecting neural activity across a broader experimental context. These two firing rate calculations address different timescales of neural activity: Eq. 3 emphasizes short-term temporal dynamics, while Eq. 4 focuses on aggregated activity over multiple trials.
>
> We have updated the manuscript to clarify the purpose of Eq. 3, ensuring a clearer distinction between these two complementary measures. Thank you for bringing this to our attention.
>
> ### **Q2.1: Configuration of E and I Connectivity**
>
> The higher "neural connectivity parameter" (max weight) for E compared to I neurons reflects a biologically plausible consideration. In biological systems, inhibitory neurons often exhibit higher firing rates than excitatory neurons, creating a natural tendency for inhibitory influence to dominate. To prevent the network from being overly suppressed and ensure it operates in an irregular spiking regime, we initialize the E-to-E weights slightly higher. This approach allows the network to maintain a dynamic balance between excitation and inhibition, fostering the emergence of complex spatiotemporal patterns.
>
> Importantly, our results demonstrate that the functional map and weight organization are self-organized and robust across a range of initial weight settings. As long as the initial weights are chosen within a reasonable range, the network consistently converges to the same functional organization. This robustness underscores the universal principles underlying the self-organization of functional maps and highlights the flexibility of the system to adapt to different initial conditions.
>
> ### **Q2.2 and Q3: Hyperparameters Relate to Empricial Findings & E-I connection stronger than E-E.**
> The empirical studies highlight the importance of maintaining controlled synaptic interactions among pyramidal cells and inhibitory (I) neurons to prevent excessive synchronization [1-2]. Specifically, the study by Holmgren et al. emphasizes that pyramidal cells within a local network often exhibit sparse and relatively weak excitatory (E) connections, ensuring that no single excitatory cell or cluster of cells dominates network activity [1]. This aligns with our approach of setting a lower learning rate for E-E connections than that of E-I to maintain diverse orientation preferences and prevent the network from falling into a synchronized state, which could otherwise limit the encoding of different orientations.
>
> Furthermore, In the study [2], the difference in synaptic strength between E-E and E-I connections is evident from the excitatory postsynaptic potential (EPSP) data shown in Fig. 2e. The study shows that the mean EPSP amplitude of E-I synapses is significantly larger compared to E-E synapses, indicating a stronger influence of inhibitory neurons on the excitatory population. Our results align with these findings, as evidenced by the synaptic weight data (see **Table b** below): the mean synaptic weight for E-E connections is 0.37 (±0.31), whereas for E-I connections, it is notably higher at 3.70 (±1.26). This greater synaptic strength at E-I connections, as reflected by both EPSP amplitudes in this study and the synaptic weight in our model, ensures effective inhibition within the network. It prevents runaway excitation by stabilizing network dynamics and maintaining functional diversity among the excitatory neurons adjusted by E-I-E loops, thereby supporting diverse orientation preferences and avoiding the full synchronization of the network.
>
> **Table b. E-E synaptic weights vs. E-I in post-trained SESNN model**
> | Synaptic Weights | Mean (±Std)   |
> |------------------|---------------|
> | EE               | 0.37 ± 0.31   |
> | EI               | 3.70 ± 1.26   |

---

> ### Author Response · Authors · 2024-11-22
> **Author Response to Reviewer zfyC (4/5)**
>
> **References:**
>
> **[1]** Holmgren, C., Harkany, T., Svennenfors, B., & Zilberter, Y. (2003). Pyramidal cell communication within local networks in layer 2/3 of rat neocortex. *Journal of Physiology, 551*(1), 139-153. https://doi.org/10.1113/jphysiol.2003.044784
>
> **[2]** Hofer, S. B., Ko, H., Pichler, B., Vogelstein, J., Ros, H., Zeng, H., & Mrsic-Flogel, T. D. (2011). Differential connectivity and response dynamics of excitatory and inhibitory neurons in visual cortex. *Nature Neuroscience, 14*(8), 1045-1052. https://doi.org/10.1038/nn.2876
>
> **[3]** Sato, T. K., Haider, B., Häusser, M., & Carandini, M. (2016). An excitatory basis for divisive normalization in visual cortex. *Nature Neuroscience, 19*(4), 568-570. https://doi.org/10.1038/nn.4250
>
>
> ### **Q4 Distinguishing Eq. 8 from the Cross-Correlogram Measure**
>
> We appreciate the reviewer’s insightful observation and the opportunity to clarify this distinction. The coefficient in Eq. 8, as employed in our study, differs fundamentally from the cross-correlogram measure described in Bair et al. (J. Neurosci., 2001), as it focuses on reliability across spike trains rather than pairwise temporal correlations.
>
> Our approach for calculating reliability is inspired by the seminal work of Haider et al. (2010, Neuron), which examines the consistency of spike trains in response to repeated presentations of the same stimulus. This measure captures the trial-to-trial reproducibility of neural responses, emphasizing the degree to which a neuron reliably encodes specific features of the stimulus. Unlike cross-correlograms, which quantify the temporal relationship between spikes from two neurons, our method aggregates reliability across multiple trials to assess the stability of a single neuron's encoding over time.
>
> We will clarify these distinctions and their importance in the revised manuscript, citing Haider et al. (2010) as a foundational reference for our method.
>
> **Reference**:
> Haider B, Krause MR, Duque A, Yu Y, Touryan J, Mazer JA, McCormick DA. Synaptic and network mechanisms of sparse and reliable visual cortical activity during nonclassical receptive field stimulation. *Neuron. 2010* Jan 14;65(1):107-21. doi: 10.1016/j.neuron.2009.12.005. PMID: 20152117; PMCID: PMC3110675.
>
> ### **Q5 Distinguishing Efficient Coding and the Role of E-E Connectivity in Functional Orientation Maps**
>
> We appreciate the reviewer’s insightful observation regarding the distinction between efficient coding and the necessity of E-E connectivity in our study. Indeed, as highlighted in the recent work by Koren et al. (eLife, 2024), efficient encoding of uncorrelated stimulus features can be achieved without E-E connectivity. In such cases, individual neurons act as independent feature detectors, relying on inhibitory (E-I) connections to decorrelate their responses and achieve sparse coding. This framework aligns with the principle of efficient coding where each neuron independently encodes features of the stimulus.
>
> However, our study emphasizes the collective dynamics of V1 as a functional map, where neurons must coordinate their activity to represent the visual field as an integrated whole. This necessitates E-E connectivity, allowing information to flow between excitatory neurons and enabling collective representation of correlated features across the visual field. The critical distinction lies in the spatial organization: in efficient coding, all neurons can access the entire stimulus, requiring only local competition via E-I connections. Conversely, in biologically realistic functional maps, each V1 neuron has a limited receptive field, necessitating E-E connections to integrate information and achieve cohesive population-level representation.
>
> We have clarified this distinction in the revised manuscript to better align our findings with the broader context of efficient coding theories and functional map dynamics.
>
> ### **Q6: Salt-and-Pepper vs. Pinwheel Maps: Trade-offs and Contextual Advantages**
> We thank the reviewer for raising this insightful point. While our results demonstrate that pinwheel maps exhibit greater robustness and efficiency in encoding static images, the prevalence of salt-and-pepper maps in many rodent species highlights the importance of evolutionary adaptations to specific ecological contexts.
>
> Firstly, rodents are primarily nocturnal and rely less on high-acuity vision compared to species with pinwheel structures, such as primates. Instead, their neural resources are more heavily allocated to somatosensory and olfactory systems, which are critical for survival in low-light environments. Secondly, the natural selection pressures on rodents prioritize wide-field vision and the ability to detect rapid motion, enabling them to evade predators. These requirements may favor the development of salt-and-pepper maps, which can provide fast processing of dynamic visual information without the need for the precision and robustness associated with pinwheel structures.

---

> ### Author Response · Authors · 2024-11-22
> **Author Response to Reviewer zfyC (5/5)**
>
> We agree that the efficiency of functional maps depends on the specific environmental and evolutionary constraints, rather than a single "most efficient" structure. We have added a discussion in the manuscript to emphasize this perspective and highlight this as an area for future investigation.
>
> **Your feedback has been invaluable in strengthening our manuscript. We have addressed your concerns thoroughly and have made significant improvements to enhance clarity and rigor. Given these revisions, we would appreciate your consideration in reassessing the evaluation score. We remain available to address any additional questions. Thank you for your thoughtful and detailed review.**

---

> > ### Comment · Reviewer_zfyC · 2024-11-25
> >
> > Dear Authors,
> >
> > I appreciated thorough and extensive revision of the paper with important improvements to the previous version. I asked three further follow-up questions. I kindly ask the Authors to reply to those, after which I will be able to finalise my score.
> >
> > Best regards

---

> > > ### Author Response · Authors · 2024-11-27
> > > **Author Response to Reviewer zfyC**
> > >
> > > Dear Reviewer zfyC,
> > >
> > > Thank you for your kind words and for acknowledging the improvements in the revised version of our paper. We appreciate your thorough review and have now addressed all three follow-up questions you raised. If you have any additional queries or concerns, we are more than happy to discuss them further.
> > >
> > > All the best,
> > >
> > > Authors

---

> > > > ### Comment · Reviewer_zfyC · 2024-11-27
> > > >
> > > > Dear Authors,
> > > >
> > > > Thank you for your thorough and comprehensive reply. I have no further questions. In light of a number of methods clarifications, text improvements and additional analyses that you provided, I raised my score for presentation, as well as the general score.

---

> > > > > ### Author Response · Authors · 2024-11-27
> > > > > **Author Response to Reviewer zfyC**
> > > > >
> > > > > Dear Reviewer zfyC, Thank you for your thoughtful review and valuable suggestions, as well as for raising the score to 8. We’re delighted our responses met your expectations and sincerely appreciate your recognition of our efforts.

---

> ### Comment · Reviewer_zfyC · 2024-11-25
>
> Thank you for including a population-level measure of reliability. It is true that single neurons in the cortex decrease variability upon stimulus presentation, but they still maintain a rather high degree of variability. The literature on predictive / efficient coding with spikes, that is highly relevant to these results but seems neglected by the authors, argues that microcircuits with predictive / efficient coding have high reliability on the level of the population-level signals, but strong variability (and thus low reliability) on the level of single neurons. One may think of an efficient microcircuit as a portion of the entire circuit of neurons modelled in your study (e.g. one pinwheel). Can authors comment on that?

---

> ### Comment · Reviewer_zfyC · 2024-11-25
>
> Comment on Q4: Thank you for elaborating on this. However, your Eq. 8 still seems to measure the correlation between spike trains x and y, as a function of the time lag m. I do not see how this is fundamentally different from the well-known measure of the cross-correlogram as described in Bair et al. 2001? Also, the R_{xy}(m) is a pair-wise measure, but authors comment that neuron's reliability is defined as a maximum cross-correlation. How can neuron's reliability (which should be a property of a single neuron) be inferred from a pair-wise measure?

---

> ### Author Response · Authors · 2024-11-27
> **Individual Neuron Reliabilty; Predictive/efficient Coding**
>
> We appreciate your follow-up question; you have raised an interesting and important issue. Clarifying the reliability of neural responses at the single-neuron level is crucial, and we provide two key references to support this discussion [1, 2]. According to the findings of studies [1] (**Fig. 2B**) and [2] (**Fig. 2a and b**), individual neurons demonstrate reliable responses to dynamic stimuli, even if their spike patterns show high variability or irregularity when examined in isolation. The reduction in across-trial variability in response to stimulus onset is a well-documented phenomenon that stabilizes the network state and reduces redundant activity, indicative of an efficient coding mechanism.
>
> In the context of predictive/efficient coding, the network exhibits high reliability at the population level, with neurons displaying irregular spiking patterns that contribute to a sparse and distributed representation. At the single-neuron level, even though their spike trains exhibit substantial variability/irregularity (as quantified by the coefficient of variation), they nonetheless maintain reliability across repeated trials. This is consistent with an efficient coding strategy, where the reliability of neural encoding is preserved despite irregular firing patterns. As shown in **Fig. 4** of our manuscript, we assessed neuron reliability by analyzing the similarity of spike trains produced under feedforward input with consistent levels of random noise. This analysis indicates that individual neurons also exhibit high reliability, in agreement with the findings of the cited references.
>
> At the population level, the variability of spike patterns (as discussed in **W5, Table a**) can be interpreted as part of an efficient coding framework. In this framework, not every neuron needs to fire for predictable stimuli; some neurons remain silent to maintain sparse coding, regulated through E-I-E loops governed by the CM rule. Specifically, when an E neuron prefers a particular orientation, it may activate a strongly connected inhibitory I neuron, which in turn suppresses neighboring E neurons that would otherwise respond to the same stimulus. Thus, the neuron can "predict" the stimulus and avoid redundant firing. Although the spike patterns across neurons may seem highly variable, this variability is structured rather than random, leading to a more reliable and efficient representation of the stimulus. In this case, each neuron maximizes its efficient coding strategy within the larger network.
>
> The pinwheel microcircuit in our model, similar to the efficient microcircuits described in predictive coding theories, demonstrates high population-level reliability by coordinating individual neuron responses to lower redundancy and improve the encoding capacity. Interactions between neurons, particularly via feedforward and inhibitory mechanisms, help maintain distinct responses while preserving adaptability to new stimuli. Consequently, individual neurons may exhibit reliability in their spike patterns, the population-level coding is highly reliable and robust against external noise as well, aligning with the theoretical principles of efficient coding.
>
>
> **References:**
>
> **[1]** de Ruyter van Steveninck, R. R., Fairhall, A. L., Lewen, G. D., & Bialek, W. (1997). Reproducibility and variability in neural spike trains. *Science, 275*(5307), 1805-1808. https://doi.org/10.1126/science.275.5307.1805
>
> **[2]** Churchland, M. M., Yu, B. M., Cunningham, J. P., et al. (2010). Stimulus onset quenches neural variability: A widespread cortical phenomenon. *Nature Neuroscience, 13*, 369-378. https://doi.org/10.1038/nn.2501

---

> ### Author Response · Authors · 2024-11-27
> **Clarifying the Calculation of Neuron's Reliability and Its Connection to Cross-Correlation**
>
> **Clarifying the Calculation of Neuron's Reliability:**  We sincerely thank the reviewer for highlighting this critical point and for allowing us to clarify the distinction and rationale behind our approach. Indeed, as the reviewer noted, Eq. 8 is mathematically similar to the cross-correlogram framework described in Bair et al. (2001), which measures the correlation between the spike trains of two neurons. As the reviewer suggested, while the mathematical principle is similar, the application of Eq. 8 differs: we use it to calculate the correlation between spike trains generated by the same neuron under identical input stimuli but with different realizations of Gaussian noise. This approach allows us to assess a neuron’s reliability in encoding the same stimulus across trials with noise variability.
>
> Having said that, we acknowledge that the methodology shares fundamental principles with Bair et al. (2001). To validate this, we directly compared our approach to the Bair et al. method under identical conditions. The following table presents the results: the first row corresponds to our method, and the second row lists results derived from the classical Bair et al. (2001) approach. The linear fit of these results ($ Y = 0.97X + 0.03 $, $ R^2 = 0.999979 $, where $Y$ is reliability calculated using the Bair method and $X$ using ours), demonstrates the equivalence of the two methods:
>
> | Noise Level ($\sigma^2$)| 0                     | 0.005                     | 0.02                     | 0.045                     | 0.08                     | 0.125                     |
> |------------------|-----------------------|-----------------------|-----------------------|-----------------------|-----------------------|-----------------------|
> | Our Method       | 1.00000010309993      | 0.377802643511030     | 0.273582933004946     | 0.253166377544403     | 0.213144703302532     | 0.139692798708424     |
> | Bair et al. (2001) | 1.00000009536743     | 0.392921480249476     | 0.290690221852726     | 0.274032339671302     | 0.242585890359349     | 0.171353012881295     |
>
> As evident from the table and the linear fit, both methods yield nearly identical results, confirming that our calculation aligns with established methodologies. We chose to adopt the Haider et al. (2010, Neuron) framework due to our group’s extensive experience with this approach, as it has been a core element of our prior work.
>
>
> **Regarding the second question** ("How can a neuron's reliability, which should be a property of a single neuron, be inferred from a pair-wise measure?"), we provide further elaboration below:
>
> Our methodology calculates reliability as the mean of pairwise correlations across multiple noise realizations, where we take the maximum value of the cross-correlation coefficient over the time lag range \([-T/2, T/2]\) as the reliability measure. For each neuron, we present a fixed stimulus with Gaussian noise of the same variance but independently sampled for each trial. The spike trains generated across these trials are compared pairwise to compute their cross-correlation coefficients, and the results are averaged across all neurons. This approach captures the robustness of a neuron’s encoding in the presence of noise, providing a quantifiable measure of reliability under varying input conditions.
>
>
> We sincerely apologize for not adequately explaining this distinction in our initial manuscript and rebuttal, which may have caused confusion. We greatly appreciate the reviewer’s insight, which has prompted us to explicitly address the relationship between our approach and classical cross-correlogram frameworks. We hope this clarification resolves the concerns.
>
> **Reference**:
>
> Haider, B., Krause, M. R., Duque, A., Yu, Y., Touryan, J., Mazer, J. A., & McCormick, D. A. (2010). Synaptic and network mechanisms of sparse and reliable visual cortical activity during nonclassical receptive field stimulation. *Neuron, 65*(1), 107–121. https://doi.org/10.1016/j.neuron.2009.12.005
>
> Bair, W., Zohary, E., & Newsome, W. T. (2001). Correlated firing in macaque visual area MT: Time scales and relationship to behavior. *Journal of Neuroscience, 21*(5), 1676–1697.

---

### Official Review · Reviewer_seTg · 2024-11-04

**Soundness:** 3
**Presentation:** 3
**Contribution:** 2
**Rating:** 6
**Confidence:** 4

**Summary:**

The paper investigates how the pattern of orientation map emerges from a recurrent spiking network model with synaptic plasticity, and concludes that the feedforward input overlap is crucial for forming different map patterns, i.e., salt-and-pepper vs. pinwheel structures.

**Strengths:**

1. Biological plausibility of the recurrent spiking network model with plasticity. The model is well supported by neurobiological experiments and is quite similar to the conventional recurrent spiking network used in neuroscience studies.

**Weaknesses:**

1. Although the authors claim they provide a spiking network mechanism of orientation map, the spiking dynamics seem unnecessary, because none of the results rely on spike timing information. All learning rules are based on the firing rate of neurons only. This makes the spiking network model look like a "strawman" a bit. I envision the spiking network model with spike-time-dependent plasticity rule might have different mechanisms from the rate-based Hebbian rules. Anyhow, I think the author can provide some explanations about how much we could gain about the orientation map formation from the proposed spiking network model.

2. I am still debating about the contribution of this work to ICLR society. That is, how much we could gain from this study to develop the next-generation AI algorithm? Or how the results in the paper can be incorporated into the modern deep learning algorithms? Another thing is although the model is biologically plausible, all learning rules and network model architecture are not new.

3. It is uncommon to see the title of the paper PDF differ from the title shown in the OpenReview. This makes me concerned that the paper might not be well prepared, at least in the submission stage.

### Minor
1. When comparing output orientation map size among different species, the model should scale with the different neuron density, RF size, and magnification factor (in Table 2) across species. For example, the patch size that each neuron perceives is a constant (16 pixel), while for different species it should be scaled with the following term:

    V1 RF size (deg) * magnification factor (mm/deg) * sqrt(neuron density) (neuron / mm)

(so the outcome is the receptive field size scaled with neuron number)

Also, the recurrent wiring size scaled with neuron number in Eq. (5), presumably conservative about 100 ~ 200 um across species, should be scaled with sqrt(neuron density) (neuron / mm) in different species.

**Questions:**

1. What is the $z_j(t)$ in Eq. 3?

2. Could the authors comment on the differences between the current model with Hansel & Vreeswijk, J. Neurosci., 2012?

---

> ### Author Response · Authors · 2024-11-22
> **Author Response to Reviewer seTg (1/7)**
>
> We appreciate the reviewer’s insightful comments regarding the role of spiking dynamics in our model and the implications for orientation map formation. Below, we provide clarifications to address these points in detail:
>
> ### **W1.1: The Role of Spike Timing Information in SESNN Model**
>
> We would like to clarify that spike-based information processing is indeed a core mechanism of the SESNN model. Specifically, the function $z_j^{\left( \mathrm{K}^{} \right)}(t) = H\left(u_j^{\left( \mathrm{K}^{} \right)}(t) - \theta_j^{\left( \mathrm{K}^{} \right)}\right)$, where $H(x)$ represents the Heaviside step function defined as $H(x) = 1$ if $x > 0$ and $0$ otherwise, governs the spike output from neuron $j$ at time $t$. Here, $z_{j}^{\left( \mathrm{K}^{} \right)}(t)$ represents whether neuron $j$ emits a spike upon reaching the threshold $\theta_j^{\left( \mathrm{K}^{*} \right)}$, clearly indicating that the SESNN model relies on spike-based processing. (Appendix A.1, line 897)
>
> Moreover, we conducted two sets of experiments to emphasize the significance of spike timing in our network:
>
> **Natural Image Encoding:** We presented the BSDS500 [1] image set to both pinwheel and salt-and-pepper organizations generated by our SESNN model. To evaluate the complexity of these spatial distributions, we used MATLAB's 'entropyfilt' function, which calculates the local entropy within a sliding window across each pixel's neighborhood. This method captures local intensity variations, effectively measuring the complexity of pixel distributions by identifying changes in spatial patterns, such as edges, corners, and other complex structures. By sliding a small window across the BSDS500 binary images, the entropy measure captures these spatial distributions, providing an effective measure of local complexity. **Using the SESNN model's spike timing properties, we were able to analyze response time differences in the encoding mechanisms between pinwheel and salt-and-pepper organizations.** This experiment indicates the function of pinwheels in detecting complex orientations (in a larger complexity value compared to those of salt-and-peppers), which improves their ability to process complex contours. Please refer to **Table a** below for details.
>
> **Table a. Neuronal response onset latency from pinwheels and salt-and-peppers relates to structural complexity.**
>
> | Response Onset Latency (ms) | Pinwheels Respond to BSDS500 Image Complexity (Normalized)  (Mean ± Std Dev) | Salt-and-Peppers Respond to BSDS500 Image Complexity (Normalized) (Mean ± Std Dev) |
> |----------------------------|----------------------------|-----------------------------------|
> | 0                          | 0.6519 ± 0.0210            | 0.4939 ± 0.0940                   |
> | 1                          | 0.5623 ± 0.0792            | 0.4900 ± 0.1016                   |
> | 2                          | 0.5309 ± 0.0756            | 0.4528 ± 0.1020                   |
> | 3                          | 0.5115 ± 0.0761            | 0.4384 ± 0.1115                   |
> | 4                          | 0.4758 ± 0.0803            | 0.4422 ± 0.1132                   |
> | 5                          | 0.4807 ± 0.1043            | 0.4330 ± 0.1051                   |
> | 6                          | 0.4632 ± 0.1178            | 0.3963 ± 0.1044                   |
> | 7                          | 0.4946 ± 0.0763            | 0.3978 ± 0.1086                   |
> | 8                          | 0.4615 ± 0.0922            | 0.3962 ± 0.0974                   |
> | 9                          | 0.4323 ± 0.1091            | 0.4134 ± 0.0921                   |
>
> **Reference:**
>
> [1] Martin, D., Fowlkes, C., Tal, D., & Malik, J. (2001). A database of human segmented natural images and its application to evaluating segmentation algorithms and measuring ecological statistics. *In Proceedings of the Eighth IEEE International Conference on Computer Vision* (Vol. 2, pp. 416–423). IEEE. https://doi.org/10.1109/ICCV.2001.937655.

---

> > ### Comment · Reviewer_seTg · 2024-11-27
> > **Reply to W.1.1**
> >
> > Thanks very much for the detailed reply.
> >
> > I do see the pinwheel structure respond quicker than the salt-and-peppers, which makes sense because pinwheel model has more recurrent interactions, and animals with pinwheel structure have better vision than the ones without. Nevertheless, my original question was asking whether the spiking activity is critical for reproducing the network model's spatial pattern (salt-and-pepper and columns). That is, if we build a rate-based neuronal network with the same learning rules, can the rate model reproduce the same spatial patterns with similar mechanisms?
> >
> > PS: My question is to distinguish whether the spiking-level mechanisms are necessary to reproduce the spatial patterns, especially the current model uses rate-based learning rules, even if the authors' rebuttal provides some analysis showing its connections with STDP (spiking version rules). This is critical for us to distinguish mechanisms and principles at different levels in the network model, rather than just simulating a model with sufficient biological details.

---

> ### Author Response · Authors · 2024-11-22
> **Author Response to Reviewer seTg (2/7)**
>
> **Spike Timing Contribution Analysis:** To further demonstrate the role of spike-based processing, we investigated the information carried by spikes in the pinwheel and salt-and-pepper structures, thus we designed a deep learning-based decoder, to reconstruct natural images directly from spike patterns. The decoder architecture ensures that temporal and spatial information carried by spikes is preserved and leveraged.
>
> Our experiments demonstrate that the spike timing information plays a critical role in decoding visual features. To highlight the temporal aspect of information encoding, we analyzed the Mean Squared Error (MSE) reduction over time during the decoding process. The results are shown in the table below:
>
> **Table b. Spike timing information plays a critical role in decoding visual features from pinwheel representations**
>
> | **Time Step** | **MSE (Mean)** | **MSE (Std)** | **PSNR (Mean)** | **PSNR (Std)** | **SSIM (Mean)** | **SSIM (Std)** |
> |---------------|----------------|---------------|-----------------|----------------|-----------------|----------------|
> | 5             | 0.23757033     | 0.03981802    | 6.30520294      | 0.75126144     | 0.04070427      | 0.02210807     |
> | 10            | 0.23136568     | 0.03776086    | 6.41733001      | 0.73466407     | 0.04538476      | 0.02925294     |
> | 15             | 0.23009922     | 0.04044247    | 6.45142324      | 0.79819216     | 0.08473944      | 0.06729780     |
> | 20             | 0.22391695     | 0.03774543    | 6.56613981      | 0.78379610     | 0.10526266      | 0.07706107     |
> | 25             | 0.21155891     | 0.03362105    | 6.80873392      | 0.77036601     | 0.13092879      | 0.08281487     |
> | 30             | 0.19971623     | 0.03164017    | 7.06103560      | 0.79017360     | 0.15610416      | 0.09238517     |
> | 35             | 0.19076131     | 0.02972120    | 7.25899343      | 0.78480566     | 0.17703904      | 0.09295404     |
> | 40             | 0.18359858     | 0.02889217    | 7.42710457      | 0.79783575     | 0.19904305      | 0.09510222     |
> | 45             | 0.17874034     | 0.02941605    | 7.55050925      | 0.84109040     | 0.21685248      | 0.09919024     |
> | 50            | 0.17654246     | 0.02931675    | 7.60599308      | 0.85225538     | 0.22804489      | 0.10168605     |
>
>
> These findings provide quantitative evidence of the role of spike timing in decoding visual information.
>
> Through these analyses, we demonstrated that the SESNN model encodes visual stimuli via spike timing, which is critical for its function and supports the biological plausibility of the network. **The role of spike timing in the model's encoding process allows for temporal sparsity and efficient information representation, distinguishing our approach from traditional rate-based encoding methods.**
>
> ### **W1.2: The Learning Rule of Spiking Dynamics in Our Proposed Model**
> In our learning rule, the averaged firing rate of the pre- $x_j$ and post-synaptic $y_i$ neurons is equivalent to the synaptic trace used in STDP. The synaptic trace is calculated as:
>
> $$
> \tau \frac{dx}{dt} = -x + \delta,
> $$
>
> where $\delta$ represents a spike event.
>
> Thus, fundamentally, the HO rule and CM rule used in our model have a symmetric STDP form:
>
> - **HO Rule**:
> $$
> \Delta W_{ij} = \frac{1}{2} y_i \delta_j + \frac{1}{2} x_j \delta_i - z_i^2 W_{ij},
> $$
> where the last term ($-z_i^2 W_{ij}$) acts as weight decay for normalization.
>
> - **CM Rule**:
> $$
> \Delta W_{ij} = \frac{1}{2} y_i \delta_j + \frac{1}{2} x_j \delta_i - x_i x_j (1 + W_{ij}),
> $$
> where the last term ($-x_i x_j (1 + W_{ij})$) serves a similar normalization purpose by controlling the weight dynamics.
>
> To stabilize network behavior during training, weight changes are accumulated separately and applied in aggregate after every 100 image patch training samples. This strategy prevents instability and ensures convergence during the training process while maintaining the influence of spiking dynamics.
>
> These rules naturally reduce to an STDP form of the learning rule. While the model uses rate-based calculations for computational efficiency, it still inherently retains spike timing information through the dynamics of synaptic traces and their influence on plasticity.

---

> > ### Comment · Reviewer_seTg · 2024-11-27
> > **Reply**
> >
> > Thanks for the detailed reply and the results from the new experiments.
> >
> >  I think the table b is fair to say the temporal information is critical for decoding visual features but not necessarily implying the _spike_ is necessary.
> >
> > In addition, I cannot get the abbreviations such as PSNR, SSIM, which are also absent in the main text.

---

> ### Author Response · Authors · 2024-11-22
> **Author Response to Reviewer seTg (3/7)**
>
> ### **W1.3: Orientation Map Formation from Our Proposed Spiking Network Model (Updated in Revised manuscript, New Section A.6)**
>
> **1. HO Rule Learns V1 Receptive Field:** The HO rule learns receptive fields from visual images, as described by the following formula:
>
> $$\Delta FF_{im}^{(\mathrm{E} \leftarrow \mathrm{Image})} \propto y_i x_m - y_i^2 FF_{im}^{(\mathrm{E} \leftarrow \mathrm{Image})},$$
>
> where $x_m$ and $y_i$ represent the gray value of pixel $m$ and instantaneous firing rate of post-synaptic neuron $i$. This rule ensures that weights converge when:
>
>
> $$FF_{im} = \frac{\langle y_i x_m \rangle}{\langle y_i^2 \rangle} = \frac{\text{STA}_i}{\langle y_i^2 \rangle}.$$
>
> Corresponding to the normalized spike-triggered average (STA). HO rule stabilizes learning by constraining synaptic growth, resulting in biologically realistic V1-like receptive fields (RFs).
>
> **2. Retinotopic Topographic Constraints:** Each neuron encodes only a small portion of the visual field (Fig. 1a), with the small patch $\in \mathbb{R}^{H' \times W'}$ smaller than the input image $I \in \mathbb{R}^{H \times W}$. The population’s collective response is needed to represent the full stimulus, aligning with the topographic mapping from retina to cortex.
>
> **3. High-Overlap Strengthens Local Excitatory-Excitatory Connections for Orientation Map Continuity:** Excitatory connections aligns adjacent neurons’ firing activities, promoting similar orientation preferences. This requires a high overlap (Fig. 2), where correlated feedforward input initially strengthens local excitatory connections, ensuring **continuous orientation preference** and forming **iso-orientation domains** with similar orientation preferences (Fig. 3e-h).
>
> **4. CM rule's Anti-Correlation via E-I-E loops Ensures Orientation Diversity and Map Coverage:** Inhibitory neurons, through the CM rule:
> $$\Delta W_{ij}^{(\mathrm{K} \leftarrow \mathrm{K}^*)}
> \propto y_i x_j - \langle y_i \rangle \langle x_j \rangle \left( 1 + W_{ij}^{(\mathrm{K} \leftarrow \mathrm{K}^*)} \right),$$
> where $x_j$ and $y_i$ represent the instantaneous firing rates of pre- and post-synaptic neuron $j$ and $i$. The operator $\langle \cdot \rangle$ denotes the moving average value (refer to Eq. 4).
>
> At equilibrium, the CM rule converges to:
>
> $$W_{ij} = \frac{\langle y_i x_j \rangle - \langle y_i \rangle \langle x_j \rangle}{\langle y_i \rangle \langle x_j \rangle}.$$
>
> We have:
>
> $$W_{ij} \propto\langle y_i x_j \rangle - \langle y_i \rangle \langle x_j \rangle = Cov(y_i, x_j).$$
>
> This covariance shapes neural connectivity based on activity covariance, where synaptic weights are strengthened between correlated neurons and weakened between uncorrelated ones. This mechanism ensures that inhibitory neurons regulate overall activity between neighbouring E neurons through the E-I-E loops (lateral inhibition), suppressing predictable responses to maintain **sparse coding**. This also prevents excitatory neurons from excitatory connections overly responding to the same orientation, enabling preference for other orientations and **ensuring full coverage**. Over larger connection ranges, highly correlated feedforward correlations are disrupted and inhibitory neurons drive diverse responses, resulting in distinct domains with varying preferred orientations, forming **pinwheel structures** within the functional orientation map.
>
>
> ### **W2.1: Potential to Inspire Next-Generation AI and Integrate into Modern Deep Learning Algorithms**
>
> **1. Next-Generation AI:**
>
> We propose that integrating our spiking neural network into the deep learning pipeline—particularly for low-power, event-driven computations—has the potential to contribute to the development of more efficient and adaptive learning systems. Specifically:
>
> **a.** The balance of E- and I- activity, shaped by synaptic plasticity learning rules, facilitates effective visual representation while aiming to reduce power consumption.
>
> **b.** The formation of pinwheel structures appears to reflect a trade-off between wiring costs and maximizing image representation capabilities, which may contribute to enhanced computational efficiency and energy savings.
>
> **c.** The organized pinwheel arrangement is associated with reduced wiring costs (Fig. 3c), improved sparse coding (Fig. 4f), greater robustness to noise (Fig. c), and enhanced information-cost efficiency (Fig. 4e).
>
> These features suggest that our model could be well-suited for applications in low-power vision systems, robust visual processing, and energy-efficient AI hardware, offering insights that might help inform future artificial neural network designs.

---

> ### Author Response · Authors · 2024-11-22
> **Author Response to Reviewer seTg (4/7)**
>
> **2. Integrate into Modern Deep Learning Algorithms:** To demonstrate the practical relevance of our model, we explore its capability as a front-end encoder for deep spiking neural networks . Specifically, we design a five-layer convolutional and pooling-based SNN to classify images from the FashionMNIST dataset. The classification accuracy achieved using our model's spike sequences is compared with that of Poisson spike sequences, as summarized in **Table c**:
>
> **Table c: Classification accuracy achieved using our model's spike sequences compares with that of Poisson spike sequences.**
>
> | **Dataset Class**      | **Accuracy(%) of Model** | **Accuracy(%) of Poisson Encoder** |
> |-------------------------|---------------------------------------|------------------------------------------|
> | T-shirt / top           | 86.80                                | 85.40                                    |
> | Trouser                 | 97.30                                | 96.40                                    |
> | Pullover                | 83.40                                | 83.80                                    |
> | Dress                   | 91.10                                | 90.30                                    |
> | Coat                    | 83.60                                | 81.60                                    |
> | Sandal                  | 96.20                                | 96.80                                    |
> | Shirt                   | 73.20                                | 67.80                                    |
> | Sneaker                 | 95.40                                | 95.40                                    |
> | Bag                     | 97.80                                | 97.00                                    |
> | Ankle boot              | 96.10                                | 94.20                                    |
> | **Overall**             | **90.09**                            | **88.87**                                |
>
> From these results, we observe that, compared to the randomness inherent in Poisson-based encoding, the functional organization of pinwheels produces spike sequences with more structured spatiotemporal patterns. This organization encodes richer information, contributing to improved classification accuracy across most categories.
>
> Additionally, we evaluate the noise robustness of our model by introducing Gaussian noise to the input images. The results are shown in **Table d** and summarized below:
>
> **Table d: Comparison of classification performance using different spiking sequence representations as inputs.**
>
> | **Noise Level** | **Pinwheels Encoder Accuracy** | **Poisson Encoder Accuracy** | **Salt-and-Peppers Encoder Accuracy** |
> |-----------------|------------------------------|--------------------------------|---------------------------------------|
> | 0.0             | 89.47%                       | 93.14%                         | 87.92%                                |
> | 0.1             | 88.98%                       | 92.14%                         | 87.21%                                |
> | 0.2             | 88.41%                       | 82.81%                         | 86.18%                                |
> | 0.3             | 88.11%                       | 62.79%                         | 84.51%                                |
> | 0.4             | 86.22%                       | 36.61%                         | 80.52%                                |
> | 0.5             | 82.91%                       | 18.79%                         | 67.39%                                |
> | 0.6             | 70.88%                       | 12.03%                         | 38.37%                                |
> | 0.7             | 43.60%                       | 10.89%                         | 19.95%                                |
> | 0.8             | 21.20%                       | 10.50%                         | 13.00%                                |
> | 0.9             | 12.51%                       | 10.20%                         | 10.71%                                |
>
> Compared to the Poisson encoder, the functional organization inherent in pinwheel encoding provides better resistance to noise, as it retains meaningful features even in noisy environments. Moreover, the salt-and-pepper encoding also shows a degree of robustness, albeit slightly less effective than pinwheel encoding under higher noise levels.
>
> These findings highlight that the spatiotemporal regularity of the pinwheel functional maps significantly benefits both classification performance and noise resistance, suggesting their potential utility in enhancing the robustness of modern deep learning frameworks.

---

> ### Author Response · Authors · 2024-11-22
> **Author Response to Reviewer seTg (5/7)**
>
> ### **W2.2: Role of E-E Learnable Connection and Network Architecture**
> We appreciate the reviewer's observation. While the individual components, such as the learning rules and architectural features, are indeed well-established, our model integrates these elements in a novel way to achieve a biologically plausible spiking neural network capable of generating both pinwheel and salt-and-pepper structures. This unique combination enables us to explore aspects of cortical organization that previous models have not adequately addressed.
>
> Specifically:
>
> **1. Retinotopic Topographic Constraints (see W1.2 response for details).** This ensures that each RF receives only a small patch from the large natural image, unlike conventional models that receive the entire image, which does not adhere to retinotopic constraints.
>
> **2. Overlapping Visual Fields.** The overlapping visual fields impact the strength of excitatory-excitatory (E-E) connections. Feedforward (FF) input influences these E-E connections, and circuits involving FF-E-E interactions are crucial for ensuring that nearby neurons have similar orientation preferences. This means that the overlapping inputs and the E-E circuits work together to maintain consistent orientation selectivity across neurons.
>
> **3. E-E Connections Are Learnable.** Unlike conventional networks, which often disregard the role of E-E connections in spiking networks, our model emphasizes their importance. Conventional approaches focus on inhibiting uncorrelated excitatory neurons to maintain independence, while our model ensures that excitatory connections align the firing activities of adjacent neurons, promoting similar orientation preferences. This requires high overlap (Fig. 2), where correlated feedforward input strengthens local excitatory connections through neural plasticity, ensuring continuous orientation preferences and forming iso-orientation domains with consistent orientation selectivity.
>
> By integrating these components cohesively, our model advances the understanding of how pinwheel and salt-and-pepper structures emerge, offering valuable insights into orientation selectivity and the mechanisms.
>
> ### **W3: Inconsistency of Title Between OpenReview and PDF**
> Apologies for the oversight. After submitting the abstract, we had extensive discussions about the content, including the title, and refined several versions to enhance the paper's readability and quality. When submitting the final PDF at the deadline, the title did not update correctly. We will correct this inconsistency to ensure alignment between the submission and OpenReview, avoiding any confusion in the future.
>
> ### **Minor 1: Comparison of Difference Pinwheel Structure Size**
> We appreciate the reviewer's suggestion to rescale model parameters based on species-specific differences. We indeed used this method to calculate orientation maps for the macaque visual cortex (Fig. 1e). Additionally, Table 2 was used to estimate the degree of overlap among different species through Eq. 16, taking into account variations in V1 RF size, cortical magnification factors in the central visual area, and neuron density. This allowed us to evaluate overlap differences among macaques, cats, and mice (Fig. 2e), verifying our views about overlap underlying the formation of pinwheel size.
>
> **We understand that Fig. 2a-d may have caused some misunderstanding, and we would like to provide further clarification:**
>
> Our goal in Fig. 2 was specifically to establish the relationship between overlap and pinwheel size, rather than to generate precise orientation maps for each species. Our study focuses on **identifying key factors that influence the formation of pinwheel structures versus salt-and-pepper organizations in visual cortical maps**. To achieve this, we analyzed extensive anatomical data, including neuron density, RF size, cortical magnification factor, retina size, and V1 size (Tables 2 and 3). We discovered that RF density can serve as a linear classifier to differentiate animals with pinwheel structures from those with salt-and-pepper organizations (Fig. 5). However, we aimed to further determine the main factors underlying RF density.
>
> In our SESNN model, we controlled all variables except for overlap, which we treated as a controllable parameter, to test its effect on pinwheel size. By systematically varying the degree of overlap, we demonstrated its significant influence on pinwheel characteristics, quantitatively representing overlap on the x-axis and other pinwheel metrics on the y-axis (Fig. 2b-d). Our aim was not to replicate specific animal visual systems, but rather to isolate overlap as a significant factor influencing pinwheel organization. Finally, we used additional anatomical data comparing overlap across mice, macaques, and cats to validate the importance of this factor (Fig. 2e).

---

> ### Author Response · Authors · 2024-11-22
> **Author Response to Reviewer seTg (6/7)**
>
> ### **Q1: Valuable $z_j(t)$ in Eq.3 Clarification**
> Thank you for pointing out this valuable $z_j(t)$.
>
> The equation is:
>
> $$
> z_j^{(\mathrm{K}^*)}(t) = H\left(u_j^{(\mathrm{K}^*)}(t) - \theta_j^{(\mathrm{K}^*)}\right),
> $$
>
> where $H(x)$ is the Heaviside step function, defined as: $H(x) = 1$ if $x > 0$ and $0$ otherwise. Here, $z_j^{(\mathrm{K}^*)}(t)$ represents the spike output from neuron $j$ at time $t$, reaching the spike threshold $\theta_j^{(\mathrm{K}^*)}$.
>
> This equation was previously introduced in Appendix A.1, line 897. To improve the readability of the main text, we will include a more detailed explanation of this Heaviside step function in the revised version of our manuscript.
>
> ### **Q2: Differences Between the Current Model and Hansel & van Vreeswijk (J. Neurosci., 2012)**
> To clarify the differences between our SESNN model and the Hansel & van Vreeswijk (2012) model, we have summarized them in the **Table e** below:

---

> ### Author Response · Authors · 2024-11-22
> **Author Response to Reviewer seTg (7/7)**
>
> **Table e: Comparison of the SESNN model (current work) with the Hansel & van Vreeswijk (2012) model.**
>
> | **Aspect**                | **Hansel & van Vreeswijk (2012)**                                                                                                                                                   | **SESNN Model (Current Work)**                                                                                                               |
> |---------------------------|-----------------------------------------------------------------------------------------------------------------------------------------------|------------------------------------------------------------------------------------------------------------------------------------------|
> | **Focus**                | Orientation selectivity in salt-and-pepper organizations, without a functional orientation map.                                                 | Emergence and functional roles of pinwheel and salt-and-pepper organizations in V1.                                                         |
> | **Natural Image Inputs**               | Randomly distributed inputs with orientation preferences; no explicit use of natural images.                                                                | Uses 160 whitened natural images as input, preprocessed to reduce pixel correlations and enhance orientation diversity.                                         |
> | **Retinotopy**           | Not included; connections are based on random anatomical distance.                                                                             | Structured feedforward input with retinotopic alignment, emphasizing overlapping areas for orientation map formation.                                      |
> | **Neural Dynamics**      | Static excitatory-inhibitory balance drives orientation selectivity; random connectivity without plasticity.                           | Dynamic spiking neural network activity with synaptic plasticity, improving robustness and efficiency.       |
> | **Learning Rules**       | None; relies on pre-determined connectivity and random feedforward inputs.                                                                     | Includes biologically inspired Hebbian-Oja (HO) and Correlation Measuring (CM) rules for feedforward and lateral synaptic plasticity.    |
> | **Neural Connectivity**         | Random recurrent connections with no specific structure.                                                                                      | Neurons are initially arranged in a 2D lattice and connected locally via Gaussian-distributed synapses.             |
> | **Selectivity Mechanism**| Achieves selectivity via excitation-inhibition balance and random feedforward tuning curves.                                                   | Selectivity emerges from correlated input, synaptic plasticity, and competitive interactions between excitatory and inhibitory neurons.  |
> | **Results**             | Explains strong orientation selectivity in salt-and-pepper organizations without structured orientation maps.                                  | Demonstrates pinwheel maps’ advantages in robust coding, efficient image representation, and noise robustness compared to salt-and-pepper. |
> | **V1-Organization Simulations**   | Applies to rodents with salt-and-pepper V1 organization.                                                                                      | Simulates both rodent (salt-and-peppers) and primate (pinwheel structures) V1 organizations.                                                        |
> | **Model Complexity**     | Simpler model with random connections and no adaptive plasticity.                                                                              | Complex, with retinotopic alignment, dynamic neural interactions, and adaptive plasticity mechanisms.                                    |
> | **Contributions**         | Provides a theoretical understanding of salt-and-pepper orientation selectivity mechanisms.                                                    | Offers insights into mechanisms and functional advantages of pinwheel structures for visual processing and coding efficiency.                           |

---

> > ### Comment · Reviewer_seTg · 2024-11-27
> > **Reply to part (7/7)**
> >
> > Sorry for my late reply. Thanks very much for the detailed replies!
> >
> > - I don't think it is fair to say "random recurrent connections with no specific structure" in Hansel & Vresswijk (2012). See Eq. 9  and the surrounding text in that paper.
> >
> > - One thing I am not clear about is whether the network model in the present study is within excitation-inhibition balanced state (strong recurrent weights with balanced E and I) which introduces strong internally generated variability (Poisson-like). If not, could you envision whether and how the (chaotic) balanced state contributes to either the salt-and-pepper pattern or column structures? PS: this is question is just curious and it is fine without replying.

---

> ### Author Response · Authors · 2024-12-02
> **Reply to Reviewer "Reply to part (7/7)"**
>
> ### **Q1.1: "...random recurrent connections within no specific structure" in Hansel & Vresswijk (2012)**
> Thank you for your insightful feedback and for pointing out the classifcation regarding the "random recurrent connections" in Hansel & Vreeswijk (2012). Our characterization focused on the functional/orientation-based structure of recurrent connections, whereas their model explicitly uses connectivity that depends solely on anatomical distance, not on functional properties like orientation preference. We apologize if our phrasing of "no specific structure" was imprecise. We will revise it as: "Recurrent connections with spatial structure determined by anatomical distance, but random with respect to functional properties."
>
> Hansel & Vreeswijk (2012) model, as defined by Eq.9, implements a spatially structured connectivity where connection probability follows a Gaussian profile based on the physical distance between neurons. These connections are made without regard to neurons' functional properties like orientation preference. This distinction is central to their paper's argument: they demonstrate that orientation selectivity can emerge even when connections are random with respect to orientation preference, as long as the basic spatial organization of cortical connectivity is preserved.
>
> ### **Q1.2: Whether poisson-like or chaotic network contributes to salt-and-peppers or column structures**
>
> Thank you for your insightful question regarding the role of excitation-inhibition balance in our model.
>
> In the early stages of our network development and learning, the firing of neurons is highly dense in the population, with excitatory neurons playing a primary role. This phase is characterized by high excitability, where excitatory neurons drive neural plasticity and facilitate the initial learning processes. As learning progresses, however, there is a gradual enhancement of inhibitory currents, which arise through lateral inhibition or surround suppression. This shift towards stronger inhibition leads to a more balanced E-I network, ultimately resulting in a more sparse firing pattern with higher variability.
>
> Pinwheel structures, as observed in our model, exhibit stronger local excitatory currents coupled with strong inhibitory feedback, enabling a more efficient sparse coding scheme. In this configuration, excitatory neurons preferentially respond to their preferred orientations while engaging in non-linear population coding through excitatory connections. At the same time, lateral inhibition helps maintain the sparsity of neural firing, as shown in Figure 4 of our manuscript. In contrast, salt-and-pepper organizations have weaker inhibitory currents, resulting in more distributed, less sparse firing patterns and weaker spatial organization.
>
> The shift towards higher sparsity in neural firing is essential for refining the network's representational properties, particularly for encoding specific orientation maps. The role of inhibitory currents becomes increasingly important as they contribute to the specificity of these representations over time. This gradual transition from a more excitable to a balanced E-I state is crucial for optimizing network function and facilitating efficient information processing.
>
> While our current model does not explicitly simulate chaotic E-I dynamics, we recognize the potential for these dynamics to influence cortical map formation. A chaotic, balanced E-I state could introduce more variability at the population level, possibly fostering diverse response patterns and contributing to the formation of salt-and-pepper or columnar structures. Indeed, it is generally accepted that the "edge of chaos"—where systems exhibit high sensitivity to stimuli and efficient information representation—plays a key role in cortical organization and may enhance information capacity at the population level [1-2].
>
> In future work, we are interested in exploring how chaotic E-I dynamics could influence the emergence of cortical maps, especially with regard to the formation of salt-and-pepper and columnar structures. We agree with the reviewer that these dynamics are a critical area for further investigation and could offer new insights into how network variability influences cortical representation.
>
> Once again, we appreciate the reviewer’s thoughtful question, which encourages us to further explore the role of E-I balance and chaotic dynamics in shaping neural organization.
>
> **Reference:**
>
> [1] Van Vreeswijk, C., & Sompolinsky, H. (1996). Chaos in neuronal networks with balanced excitatory and inhibitory activity. _Science,  274_(5293), 1724-1726. https://doi.org/10.1126/science.274.5293.1724
>
> [2] Toyoizumi, T., & Abbott, L. F. (2011). Beyond the edge of chaos: Amplification and temporal integration by recurrent networks in the chaotic regime. _Physical Review E: Statistical, Nonlinear, and Soft Matter Physics, 84_(5), 051908. https://doi.org/10.1103/PhysRevE.84.051908

---

> ### Author Response · Authors · 2024-12-02
> **Response to Reviewer "Reply to W.1.1": Spiking vs. Rate-Based Mechanisms for Spatial Pattern Formation  (1/3)**
>
> ### **Q2.1: Regarding pinwheel structure and salt-and-peppers respond to different structural complexity of natural images.**
> Thank you for your comment. It seems there might be a misunderstanding regarding our new experiment on neuronal response onset latency in pinwheel and salt-and-pepper structures.
>
> Our experiment aims to investigate how the structural complexity of these organizations relates to their ability to distinguish different textures in natural images. Specifically, we observe that pinwheel structures, with their more organized and recurrent connectivity, tend to respond faster to complex contours in the image, reflecting their ability to process intricate visual features earlier than simpler orientations. On the other hand, salt-and-pepper organizations, which are more randomly distributed, do not show the same sensitivity to complex contours and tend to respond later, primarily detecting simpler orientations.
>
> Thus, the response onset latency serves as an indicator of how well these structures can distinguish complex versus simple textures in natural images. Pinwheel structures show a marked advantage in detecting complex features earlier, whereas salt-and-pepper organizations do not exhibit the same capability. This temporal information is critical for accurate encoding, as we demonstrated in the additional experiment (Table C, 4/7 of our answer), where scrambling spike timing resulted in a significant drop in classification accuracy.
>
> Rate-based models typically focus on the overall firing rate of neurons, which makes them less sensitive to precise temporal dynamics, such as the faster onset of responses to complex stimuli in pinwheel structures. These models would not directly reveal how pinwheel networks respond first to complex features, while salt-and-pepper networks lag in response. As a result, rate-based models might not be as effective at capturing the differential response timing seen in structures like pinwheels and salt-and-peppers in our new experiment. We provide further comparisons between our spiking model and rate-based models in our response to your next question.

---

> ### Author Response · Authors · 2024-12-02
> **Response to Reviewer "Reply to W.1.1": Spiking vs. Rate-Based Mechanisms for Spatial Pattern Formation (2/3)**
>
> ### **Q2.2: Some rate-based model reproduce spatial patterns.**
> Thank you for your thoughtful question.
>
> Yes, rate-based models can indeed reproduce similar spatial patterns, such as columnar structures. While spiking neural networks naturally rely on precise spike timing and event-driven dynamics, rate-based models capture key aspects of the spatial organization by focusing on the population activity and overall synaptic dynamics.
>
> For instance, Stevens et al. (2013) [1] demonstrate how a rate-based model can produce robust orientation maps in V1, showing that contrast-gain control in the early visual pathway and homeostatic adaptation in V1 neurons. Gain control regulates incoming signals by compressing varying input contrasts into a manageable range, while homeostatic adaptation allows V1 neurons to maintain stable activity levels by adjusting their firing thresholds. Through Hebbian learning regulated by these mechanisms, neurons that start with rough retinotopic mapping gradually develop orientation selectivity and organize into the characteristic "pinwheel" patterns. Lufkin et al. (2022) [2] uses a grid of excitatory neurons coupled with inhibitory neurons, where each excitatory neuron responds to specific edge orientations in visual input. Through local connections, neurons excite their immediate neighbors while inhibiting those slightly further away. This creates a dynamic system governed by competing forces: direct response to visual features, local excitation encouraging similar responses among nearby cells (lateral facilitation), and broader connections preventing excessive activation (lateral Inhibition). When trained on natural images, these mechanisms cause neurons with similar orientation preferences to cluster together while maintaining smooth transitions between different orientations. In addition to spiking and rate-based models, another approach for reproducing functional organization in visual cortex is demonstrated in the work of Margalit et al. (2020) [3], where they introduce a deep artificial neural network framework. Although this model is not spiking in nature, it incorporates a SimCLR Loss along with local spatial constraints to successfully reproduce the spatial arrangement found in V1, including the development of orientation maps. Their approach shows that even without the temporal dynamics inherent in spiking neural networks, ANN-based models with proper constraints can also produce spatial organization similar to biological systems, demonstrating that certain topological features of the visual cortex, such as orientation maps and receptive field arrangements, can emerge from a variety of modeling approaches.
>
> The fundamental distinction between spiking and rate models in orientation map formation reveals how temporal precision enables deeper mechanistic insights. The spiking model employs dual learning mechanisms (Hebbian-Oja for receptive fields and Correlation Measuring through E-I-E loops for map diversity) while crucially preserving precise spike timing information. This temporal precision allows the model to reveal distinct spiking patterns between pinwheels and salt-and-pepper organizations, providing unique insights into the computational and metabolic constraints of different cortical architectures. Specifically, we can analyze wiring costs (Figs. 3c and 4e) and energy efficiency at the level of spikes from salt-and-peppers to pinwheels (see **Table f**), and understand how spike timing carries additional information (We explained detailedly in our answer to your next question, see thread **Reply to Reviewer "Reply" (1/2)**) about neural computation that is lost in rate-based abstractions. In contrast, while rate models like GCAL [1] can reproduce similar map patterns through mechanisms like homeostatic adaptation and gain control, they necessarily abstract away these temporal dynamics and therefore cannot capture these deeper insights into the biological efficiency and information processing capabilities of different cortical organizations. This makes the spiking model particularly valuable for understanding not just how maps form, but why certain organizational principles may have evolved from a computational and metabolic perspective.
>
> **Table f. Entropy of weight distribution and entropy of neural activity.**
> | **V1 Organizations** |**Entropy of Weight Distribution (bits)** | **Entropy of Neural Activity (bits)** |
> |--|-----------------------------------|---------------------------------|
> |Salt-and-peppers  |5.5093             | 47.6400                         |
> |Pinwheels         |6.8373             | 63.0911                         |

---

> ### Author Response · Authors · 2024-12-02
> **Response to Reviewer "Reply to W.1.1": Spiking vs. Rate-Based Mechanisms for Spatial Pattern Formation (3/3)**
>
> **Reference:**
>
> [1] Stevens, J.-L. R., Law, J. S., Antolík, J., & Bednar, J. A. (2013). Mechanisms for stable, robust, and adaptive development of orientation maps in the primary visual cortex. _Journal of Neuroscience, 33_(40), 15747–15766. https://doi.org/10.1523/JNEUROSCI.1037-13.2013
>
> [2] Lufkin, L., Puri, A., Song, G., Zhong, X., & Lafferty, J. (2022). Emergent organization of receptive fields in networks of excitatory and inhibitory neurons. _arXiv_ preprint arXiv:2205.13614. https://arxiv.org/abs/2205.13614
>
> [3] Margalit, E., et al. (2020). A unifying framework for functional organization in early and higher ventral visual cortex. _Neuron, 112_(14), 2435–2451.e7. https://doi.org/10.1016/j.neuron.2020.10.011

---

> ### Author Response · Authors · 2024-12-02
> **Reply to Reviewer "Reply" (1/2)**
>
> We apologize for the delayed response. In the meantime, we took advantage of the rebuttal-discussion period to conduct additional experiments to further demonstrate that spike timing plays a crucial role in our model. We appreciate the reviewer’s insightful comments on rate-coding and spike coding, which remain core topics in both neuroscience and artificial intelligence. The energy-efficient and high-performance nature of spike coding has significant implications for developing the next generation of AI systems.
>
> To provide a fair comparison with rate-based models, we introduced two additional control experiments:
>
> 1. **Rate A**: We classified using the firing rate extracted from spike trains. Specifically, we smoothed the spike sequence over a 25-step time window and converted it into a firing rate, which was then fed into a classifier with the same architecture as our spiking model. We performed this classification on the Fashion MNIST dataset and evaluated performance under different noise conditions.
>
> 2. **Rate B**: We mathematically designed a rate-based model that mimics the architecture of our spiking model. The receptive fields of each neuron were sampled from a Gabor filter based on the neuron’s preferred orientation, matching the orientation preferences in our spiking model. The forward stimuli were first processed through the Gabor-filter network and then converted to firing rates using a tanh-shaped F-I curve with a threshold. These firing rates were then input into a similar classifier for classification. This model is designed to have the same pinwheel-like functional architecture as our spiking model.
>
> The experimental results are summarized in **Table g**, which shows the classification performance under different noise level.
>
> **Table g: Comparison of classification performance between spiking sequences and firing rates.**
>
> | **Noise Level** | **Pinwheel Spiking Sequence** | **Rate A Accuracy** | **Rate B Accuracy** |
> |-----------------|------------------------------|---------------------|---------------------|
> | 0.0             | 89.47%                       | 89.16%              | 86.17%              |
> | 0.1             | 88.98%                       | 88.91%              | 86.13%              |
> | 0.2             | 88.41%                       | 88.08%              | 85.44%              |
> | 0.3             | 88.11%                       | 85.94%              | 85.57%              |
> | 0.4             | 86.22%                       | 75.45%              | 85.08%              |
> | 0.5             | 82.91%                       | 50.81%              | 80.61%              |
> | 0.6             | 70.88%                       | 27.83%              | 49.22%              |
> | 0.7             | 43.60%                       | 17.13%              | 21.23%              |
> | 0.8             | 21.20%                       | 12.06%              | 13.75%              |
> | 0.9             | 12.51%                       | 10.00%              | 11.05%              |
>
> From this comparison, it is evident that the Pinwheel Spiking Sequence model exhibits the best performance. While all three models achieve high classification accuracy under low noise conditions, this indicates that, in the absence of noise, both spike timing and spike rate contribute effectively to the faithful representation of the image features. However, as the noise level increases, noise-induced perturbations, such as spurious spikes or missed spikes, degrade the accuracy of rate-based encoding. In contrast, the temporal information encoded in the precise spike timing remains robust, allowing the Pinwheel Spiking Sequence to maintain higher classification accuracy despite the added noise. This suggests that the temporal order of spikes plays a critical role in preserving the encoded visual features, offering additional discriminative power in noisy environments.
>
> Our findings highlight the importance of spike timing information in our model, especially as noise levels rise. While both Rate A and Rate B models experience significant performance degradation, the Pinwheel Spiking Sequence model demonstrates that spike timing—rather than spike rate alone—plays an essential role in encoding visual features with greater accuracy, particularly under noisy conditions. This underscores the potential of spike timing as a crucial factor in achieving high-fidelity visual feature encoding, which is vital for the robustness of neural coding in real-world scenarios.

---

> ### Author Response · Authors · 2024-12-02
> **Reply to Reviewer "Reply" (2/2)**
>
> We appreciate the reviewer’s valuable feedback regarding the absence of definitions for PSNR and SSIM in our previous response. We apologize for this oversight and would like to provide a detailed explanation of these metrics.
>
> PSNR (Peak Signal-to-Noise Ratio) and SSIM (Structural Similarity Index), along with MSE (Mean Squared Error), are commonly used metrics for evaluating image reconstruction quality. These metrics assess different aspects of image fidelity, providing complementary insights into the effectiveness of image reconstruction or denoising processes.
>
> ### **PSNR (Peak Signal-to-Noise Ratio)**
>
> PSNR is a traditional metric used to measure the quality of a reconstructed image compared to the original. It is based on the ratio of the maximum possible pixel value to the Mean Squared Error (MSE) between the original and the reconstructed image. PSNR focuses primarily on the fidelity of pixel values and is particularly useful for evaluating pixel-based distortions.
>
> The formula for PSNR is:
>
> $$
> PSNR = 10 \times \log_{10}\left(\frac{MAX_I^2}{MSE}\right)
> $$
>
> Where:
> - $MAX_I$ is the maximum possible pixel value (e.g., 255 for 8-bit images),
> - $MSE$ is the Mean Squared Error between the original image $I$ and the reconstructed image $I'$.
>
> ### **SSIM (Structural Similarity Index)**
>
> In contrast to PSNR, SSIM is a more perceptually relevant metric that evaluates the structural similarity between two images. SSIM considers not only pixel values but also luminance, contrast, and structural patterns. It compares local patterns of pixel intensities that are normalized for luminance and contrast, capturing how much the structure of the image has been preserved. SSIM better reflects human perception of image quality.
>
> The formula for SSIM between two images \(I\) and \(I'\) is:
>
> $
> SSIM(I, I') = \frac{(2\mu_I \mu_{I'} + C_1)(2\sigma_{II'} + C_2)}{(\mu_I^2 + \mu_{I'}^2 + C_1)(\sigma_I^2 + \sigma_{I'}^2 + C_2)}
> $
>
> Where:
> - $\mu_I$ and $\mu_{I'}$ are the mean intensities of the images $I$ and $I'$,
> - $\sigma_I^2$ and $\sigma_{I'}^2$ are the variances of $I$ and $I'$,
> - $\sigma_{II'}$ is the covariance between $I$ and $I'$,
> - $C_1$ and $C_2$ are constants used to stabilize the division with weak denominators (typically set as $C_1 = 10^{-4}$ and $C_2 = 10^{-4}$).
>
> In our experiments, we utilize both MSE,PSNR and SSIM to evaluate the quality of reconstructed images as a measure of the encoding capacity of different functional organizations. Higher PSNR and SSIM values indicate better preservation of the original visual features and structural integrity, allowing us to assess the efficiency of the encoding mechanisms of different functional organizations.
>
> **Thank you again for your valuable feedback, which has helped us strengthen our argument.**

---

> ### Author Response · Authors · 2024-12-02
> **Author Response to Reviewer seTg**
>
> We appreciate the opportunity to enhance our paper and are committed to delivering a revised version that meets high standards of clarity and thoroughness. We kindly ask that you reconsider the score in light of our clarifications. If you have further questions, we remain available to provide additional explanations.

---

> > ### Comment · Reviewer_seTg · 2024-12-02
> >
> > I appreciate the authors' detailed reply, especially the comparisons and explanations between the spiking model and the rate model. Therefore I increase my overall rating from 5 to 6. I encourage the authors to incorporate our discussions into the revised manuscript.

---

> > > ### Author Response · Authors · 2024-12-03
> > > **Author Response to Reviewer seTg**
> > >
> > > We sincerely thank you for reconsidering our work and for raising the score. We greatly appreciate your engagement with our work and your encouragement to include the discussions in the revised version. We will make sure to incorporate these valuable discussions into the manuscript to benefit future readers.

---

### Author Response · Authors · 2024-11-22
**General Response (2/3)**

### **General Response 2: Distinguishing SESNN from Other Models by Pinwheel Maps and Functional Advantages**

**Retinotopic Constraints, Overlapping Visual Fields, and Learnable E-E Connections:** Our model emphasizes the biological fidelity of retinotopic topographic constraints, ensuring each receptive field processes only a small local patch from natural images, unlike conventional models that process entire images without retinotopic constraints. Overlapping visual fields play a crucial role in maintaining consistent orientation selectivity among neurons, which is essential for analyzing the influence of feedforward heterogeneity and the resulting orientation map formation (Fig. 2). Another important feature of the SESNN model is its learnable excitatory-excitatory (E-E) connections. Unlike other models that typically focus on excitatory-inhibitory dynamics, our model demonstrates that excitatory neurons can enhance orientation selectivity through local plasticity, effectively aligning adjacent neurons' firing activities. These learnable E-E connections promote orientation similarity and continuous orientation preference, resulting in iso-orientation domains, distiguishing pinwheel and salt-and-pepper patterns.

**Natural Image Encoding & Spike Timing Contribution:** The SESNN model encodes visual information through spike timing, offering a key distinction from traditional rate-based models. We conducted the new experiment using the natural image set to show that the spike timing of pinwheels provides a more complex and efficient encoding of natural images compared to salt-and-pepper structures (**Reviewer seTg, W1.1, Tables a and b**). This temporal precision improves robustness and enhances coding efficiency, supporting the biological plausibility of our network and emphasizing the functional advantages of pinwheel structures.

**Comparative Analysis with Existing Models:** Our SESNN model has several advantages compared to previous models (Hansel & van Vreeswijk, Stevens et al.) (**Reviewer seTg, Q2, Table c & Reviewer GMwp, W1, Table a**). Specifically, our model goes beyond traditional salt-and-pepper organizations, showcasing the emergence of functional pinwheel maps, which provide functionally enhanced noise robustness, computational efficiency with trade-off between wiring costs and representation capabilities. Moreover, the SESNN model incorporates dynamic spiking activity with synaptic plasticity, whereas earlier models often relied on pre-determined connections or random anatomical distances.

### **General Response 3: Integration of SESNN Orientation Maps into Modern Deep Learning Algorithms**

Our SESNN model demonstrates its utility as a front-end encoder for deep learning, particularly in image classification tasks. Compared to Poisson spike encoding, our model's pinwheel organization produces structured spatiotemporal patterns that significantly enhance classification accuracy **(Reviewer seTg, W2.1, Table c)**. Furthermore, our evaluations under noisy conditions reveal that the pinwheel-based encoder exhibits superior robustness, effectively retaining meaningful features even as noise levels increase **(Reviewer seTg, W2.1, Table d)**. These results suggest that integrating our biologically inspired SESNN model into deep learning pipelines can improve both performance and resilience, making it valuable for advancing noise-resistant AI systems.

We appreciate **Reviewer seTg's (W2.1)** feedback, highlighting that integrating our SESNN orientation map into deep learning algorithms could make a meaningful contribution to the AI community.

### **General Response 4: Refinement of Paper Layout, Typographical Errors, Text Clarity; New Additional Experiments**

We appreciate the **Reviewer zfyC**'s detailed comments on **refining** our paper's **layout**, **correcting typographical errors**, and **enhancing text clarity**. We have thoroughly reviewed the manuscript and implemented them in the updated manuscript **using red text**.

In response to the reviewers' comments, we added **new experiments** to validate our findings further:

**1.** Neuronal response onset latency from pinwheels and salt-and-peppers relates to structural complexity. **(Reviewer seTg, Table a)**

**2.** Spike timing information plays a critical role in decoding visual features. **(Reviewer seTg, Table b)**

**3.** Classification accuracy achieved using our model's spike sequences compares with that of Poisson spike sequences. **(Reviewer seTg , Table c; Reviewer GMwp, Table b)**

**4.** Comparison of classification performance using different spiking sequence representations as inputs. **(Reviewer seTg , Table d; Reviewer GMwp, Table c)**

**5.** Representational stability for both pinwheel and salt-and-pepper organizations. **(Reviewer zfyC, Table a)**

---

### Author Response · Authors · 2024-11-22
**General Response (1/3)**

We sincerely thank the reviewers for their valuable suggestions and insightful comments, which have significantly improved our work. These inputs have helped us better explain the mechanisms underlying the formation of two distinct functional maps in V1—pinwheel structures and salt-and-peppers—through a spiking neural network model with biologically plausible synaptic plasticity, emphasizing their differences in encoding efficiency and robustness. Throughout the rebuttal-discussion period, we have addressed each point raised and clarified the presentation of our work. **The revised manuscript is now available.**

We have outlined our timeline for submitting the final revised manuscript (please refer **table below**). The revisions address your valuable feedback and aim to strengthen the clarity and quality of our work. Should you find these improvements satisfactory, we would appreciate your consideration in updating your reviewer score and summary accordingly. We sincerely thank you for your time and thorough review.

| Date       | Task  |
|------------|-------|
| 11/23      | First Revision by Authors *(the **PDF** button (top-right))*|
| 11/23-25   | Feedback from reviewers |
| 11/26      | Final Revision by Authors, merged into `SESNN_final_manuscript.pdf`|
| 11/27      | Final update of reviewer scores and summaries |


### **General Response 1: Orientation Map Formation in Our Proposed Spiking Network Model (Updated in Revised manuscript, New Section A.6)**

**Role of HO Rule: Learning Receptive Fields.** This SESNN model uses a combination of Hebbian learning (through the Hebbian-Oja (HO) rule) and anti-correlation (through the Correlation Measuring (CM) rule from the E-I-E loop) to develop biologically realistic visual maps in the cortex. The HO rule is used to teach the spiking neural network model how to form receptive fields. Receptive fields are the regions of the visual field that individual neurons respond to. The model learns these receptive fields from visual images using a learning rule based on the correlation between the input and the activity of neurons.

**Retinotopic Topographic Constraints.** The model also respects the retinotopic organization of the visual cortex. In biology, retinotopy means that adjacent neurons in the cortex correspond to adjacent regions in the visual field. The spiking model enforces this constraint by ensuring that each neuron in the network is only exposed to a localized patch of the input image, rather than the entire visual field.

**High Overlap Enhances Local Excitatory Connections for Map Continuity.** The excitatory-excitatory connections between neurons are learned in a way that strengthens the connections between neurons with overlapping receptive fields. This means that neurons that "see" similar parts of the visual field form strong local connections with each other. This concept of overlapping receptive fields is critical because it ensures that neurons next to each other have similar orientation preferences, leading to a smoothly varying iso-orientation domain.

**Role of CM Rule: Anti-Correlation via E-I-E loops Maintains Diversity and Full Map Coverage.** Meanwhile, the CM rule is used to promote diversity and coverage in the orientation map. Detailedly, the CM rule ensures that excitatory neurons do not become overly correlated through E-I-E loops, promoting diversity and full coverage of orientation preferences across the cortical map.

**Pinwheel Map Formation.** Together, these mechanisms help in forming biologically plausible pinwheel structures and iso-orientation domains, which are crucial for understanding how the visual cortex processes complex visual stimuli.

We appreciate the feedback from all reviewers, and are eager to provide insights into the formation of the pinwheel orientation map from our SESNN perspective. Detailed explanations can be found in our responses to **Reviewer seTg (W1.2)** and **Reviewer GMwp (Q1.1)**. Additionally, the discussion on the functional roles of HO and CM learning rules may also help address **Reviewer zfyC's (W3)** question.

---

### Author Response · Authors · 2024-11-25
**Quick Response**

We sincerely appreciate all the reviewers' valuable suggestions. Many of these have been incorporated into the revised manuscript (accessible via **the PDF button at the top-right**), including improvements in **layout**, **error correction**, **enhanced text clarity**, and **reducing acronyms** to improve readability. To further clarify **the rationale behind using the two Hebbian-like HO and CM rules** (see revised manuscript, lines 143-153), **the function of Eqs. 3 and 4** (lines 180-183), and **the process of orientation map formation in our proposed spiking network model**, we have added a new Section A.6 in revised manuscript, along with other minor revisions. All changes are highlighted **in red**.

We gratefully await your valuable feedback to make our paper even stronger!

---

### Author Response · Authors · 2024-11-27
**Gratitude for Score Revisions**

Dear Reviewers seTg, zfyC, and GMwp, we are thrilled to see your **scores raised from 5 to 6 and from 6 to 8**. Thank you once again for your invaluable feedback and support in enhancing our paper. We truly appreciate your constructive insights.

---

### Author Response · Authors · 2024-12-02
**General Response (3/3)：A New Additional Experiment**

In response to Reviewer seTg's insightful comments, we conducted an additional experiment to further validate our findings. Specifically, we compared the spiking sequences generated by our Pinwheel Model with firing rate models derived through two distinct methods (**Reviewer seTg, Table g**). These comparisons highlight the critical role of spike timing in encoding visual information effectively.

---

### Meta-Review · Area_Chair_Z6Zn · 2024-12-16

**Metareview:**

This paper explores the computational impact of structured topological maps in neural processing. The authors present a model of sensory cortex wherein Hebbian plasticity and physiological parameters are combined to shape the network connections. This results in topological maps that resemble those seen in cortex: random "salt-and-pepper" for small networks, large, "pinwheel" structures for big networks. The authors claim that critical factors, such as the degree of input visual field overlap, neuronal density, and the balance between localized versus and long-range connectivity competition determine the which type of topolgy emerges. Furthermore, the authors claim that the "large mammal" like pinwheel structures exhibit lower wiring cost and enhanced sparse coding capabilities compared to salt-and-pepper organizations, and that they maintain greater coding robustness against noise.

The strengths of this paper are that it well-connected to experimental data and well-motivated by it, and it brings potentially important insights into why real brains exhibit a continuum between salt-and-pepper and pinwheel structures. It largely supports its empirical claims as well through its experiments. The weaknesses are that the background literature review was limited and its interest for the ICLR community specifically is not very clear (it is really more of a computational neuroscience paper).

On balance, given these considerations and the scores (average of 7.33), a decision of accept (poster) was reached.

**Additional Comments On Reviewer Discussion:**

The authors and reviewers engaged in good discussion, and the authors attempted to address the main concerns. In particular, their results showing improved accuracy when incorporated into deep neural networks (relative to standard Poisson spiking - Tables c & d of rebuttal) is important for this paper being relevant to the ICLR audience. This data needs to be in the revised paper.

---

### Decision · Program_Chairs · 2025-01-22

Accept (Spotlight)